# Airborne observation during KORUS-AQ show aerosol optical depths are more spatially self-consistent than aerosol intensive properties

Samuel E. LeBlanc[1,2], Michal Segal-Rozenhaimer[1,2,3], Jens Redemann[4], Connor Flynn[4], Roy R. Johnson[2], Stephen E. Dunagan[2], Robert Dahlgren[2,5], Jhoon Kim[6], Myungje Choi[6,7,8], Arlindo da Silva[8], Patricia Castellanos[8], Qian Tan[1,2], Luke Ziemba[9], Kenneth Lee Thornhill[9,10], Meloë Kacenelenbogen[8]

[1]Bay Area Environmental Research Institute (BAERI), Moffett Field, CA, 94035, USA
[2]NASA Ames Research Center, Moffett Field, CA, 94035, USA
[3]Department of Geophysics, Porter School of the Environment and Earth Sciences, Tel-Aviv University, Tel Aviv, 6997801, Israel
[4]University of Oklahoma, School of Meteorology, Norman, OK, 73019, USA
[5]California State University, Monterey Bay, Seaside, CA, 93955, USA
[6]Yonsei University, 50 Yonsei-ro Seodaemun-gu, Seoul, 03722, Republic of Korea
[7]Joint Center for Earth Systems Technology (JCET), University of Maryland, Baltimore County (UMBC), Baltimore, MD, 21250, USA
[8]NASA Goddard Space Flight Center, Greenbelt, MD, 20771, USA
[9]NASA Langley Research Center, Hampton, VA, 23666, USA
[10]Science Systems and Application Inc., Hampton, VA, 23666, USA

*Correspondence to*: Samuel E. LeBlanc (samuel.leblanc@nasa.gov)

## Abstract

Aerosol particles can be emitted, transported, removed, or transformed, leading to aerosol variability at scales impacting the climate (days to years and over hundreds of kilometers) or the air quality (hours to days and from meters to hundreds of kilometers). We present the temporal and spatial scales of changes in AOD (Aerosol Optical Depth), and aerosol size (using Ångström Exponent; AE, and Fine-Mode-Fraction; FMF) over Korea during the 2016 KORUS-AQ (KORea-US Air Quality) atmospheric experiment. We use measurements and retrievals of aerosol optical properties from airborne instruments for remote sensing (4STAR; Spectrometers for Sky-Scanning Sun Tracking Atmospheric Research) and in situ (LARGE; NASA Langley Aerosol Research Group Experiment) on board the NASA DC-8, geostationary satellite (GOCI; Geostationary Ocean Color Imager; Yonsei aerosol retrieval (YAER) version 2) and reanalysis (MERRA-2; Modern-Era Retrospective Analysis for Research and Applications, version 2). Measurements from 4STAR when flying below 1000 m, show an average AOD at 501 nm of 0.36 and an average AE of 1.11 with large standard deviation (0.12 and 0.15 for AOD and AE respectively) likely due to mixing of different aerosol types (fine and coarse mode). The majority of AODs due to fine mode aerosol is observed at altitudes lower than 2 km. Even though there are large variations, for 18 out of the 20 flight days, the column AOD measurements by 4STAR along the NASA DC-8

flight trajectories matches the south-Korean regional average derived from GOCI. GOCI-derived FMF, which was found to be slightly low compared to AERONET sites (Choi et al., 2018), is lower than 4STAR's observations during KORUS-AQ.

Understanding the variability of aerosols helps reduce uncertainties in aerosol direct radiative effect by quantifying the errors due to interpolating between sparse aerosol observations sites or modeled pixels, potentially reducing uncertainties in the upcoming observational capabilities. We observed that, contrary to prevalent understanding, AE and FMF are more spatially variable than AOD during KORUS-AQ, even when accounting for potential sampling biases by using Monte

Carlo resampling. Averaging between measurements and model for the entire KORUS-AQ period, a reduction in correlation by 15% is 65.0 km for AOD and shorter at 22.7 km for AE. While there are observational and model differences, the predominant factor influencing spatial-temporal homogeneity is the meteorological period. High spatio-temporal variability occurs during the dynamic period (25–31 May), and low spatio-temporal variability during blocking pattern (01-07

June). While AOD and FMF/AE are interrelated, the spatial variability and relative variability of these parameters in this study indicate that microphysical processes vary at shorter scales than aerosol concentration processes. Where, microphysical processes such as aerosol particle formation, growth, and coagulation mostly impact the dominant aerosol size (characterized by e.g., FMF/AE) and to some degree AOD. In addition to impacting aerosol size, aerosol concentration

processes such as aerosol emission, transport, and removal mostly impact the AOD.

## Plain Language (Short) Summary

Airborne observations of atmospheric particles and pollution over Korea during a field campaign in May-June 2016 showed that the smallest atmospheric particles are present in the lowest 2 km of the atmosphere. The aerosol size is more spatially variable than their optical thickness. We show

this with remote sensing (4STAR), in-situ (LARGE) observations, satellite measurements (GOCI), and modeled properties (MERRA-2), and it is contrary to current understanding.

## Keywords

KORUS-AQ; Aerosol Optical Depth; airborne observations; aerosol intensive properties

## 1 Introduction

Aerosol interactions with light are governed by their intensive and extensive properties (Rajesh and Ramachandran, 2020). Intensive properties represent the aerosol optical properties that do not scale with aerosol concentration or mass, such as Ångström exponent (AE), fine mode fraction (FMF), single scattering albedo, asymmetry parameter, index of refraction, and hemispheric backscatter fraction. These intensive properties depend on the intrinsic properties of the aerosol; its size, shape and composition (Russell et al., 2010) and can capture the dominant aerosol speciation

(e.g., organic aerosols, black carbon, sulfate, nitrate, ammonium, dust or sea salt) (Kacenelenbogen et al., 2022). Conversely, extensive properties such as aerosol optical depth (AOD), extinction, scattering, and absorption are predominantly dependent on the amount of aerosol particles present.

The spatio-temporal scales at which the extensive and intensive properties vary are directly linked to the processes governing the emission, transport, removal, and transformation of the aerosol particles. The modelled aerosol lifetime

or rate of change is directly represented by these aerosol processes (e.g., Tsigardis et al., 2014; Hodzic et al., 2016; Saide et al., 2020), however, assessing the aerosol processes using atmospheric observations alone (i.e., without any model simulations) require high spatial and temporal resolution observations of multiple aerosol intensive and extensive properties. Anderson et al. (2003) described the use of aerosol's variability across spatial and time scales, as observed by multiple measurements, which is needed for the integration used to calculate aerosol radiative forcing

of climate, thus reducing its uncertainties. The high aerosol variability across scales shorter than 200 km in a few locations was attributed to the aerosol processes such as patchy sources, sinks, and short residence time of tropospheric particles. Shinozuka and Redemann (2011) showcased the difference between high aerosol spatio-temporal variability in the Arctic, with freshly emitted biomass burning plumes from boreal forest fire, and low variability for aerosol long-distance transport near the north pole. Targino et al. (2005) presented cases where the aerosol extensive properties

(scattering and absorption coefficients) change at scales smaller than the airmass/mesoscale, compared to intensive properties (AE and single scattering albedo) that varied much less.

Understanding the variability of aerosols helps reduce uncertainties in aerosol direct radiative effect by quantifying the errors due to interpolating between sparse aerosol observations sites or modeled pixels. Since the aerosol direct radiative effect is an integral over time and space, variations of aerosol impact its derivation, e.g., integrating the

radiative effect over 20 km for long distance aerosol transport in the Canadian Arctic will only be subject to variations of 2%, while integrating over aerosol from boreal forest fires will give 19% variations (Shinozuka and Redemann, 2011; Kacenelenbogen et al., 2019). This impact can be important not only due to aerosol variability but also due to changes in underlying surface, like when aerosol overly clouds or varying surface (from sea to land). Min and Zhang (2014) presented potential biases in the direct aerosol radiative effect by up 10% when using mean gridded values as

input, which are susceptible to subgrid horizontal heterogeneity. Defining the length scales at which aerosol variability impacts quantification of the direct aerosol radiative effect is important for reducing uncertainties in the upcoming observational capabilities afforded by the AOS (Atmosphere Observing System).

Similarly, quantifying the scale of aerosol variability can reduce errors in modelling atmospheric particles impacting air quality where observations are sparse and horizontal variations of these small aerosol particles often occur at scales
shorter than the spacing between observations. Additionally, constraining the scales at which natural variations occur is necessary to quantify the minimum collocation criteria when combining multiple observations and model platforms from AOS to upcoming field campaigns. This is similar to the question "How Long Is Too Long?" described by Sayer (2020) or for suborbital validation plans for the upcoming GEMS satellite (Park et al., 2020b). In addition, retrieval biases like small scale cloud contamination or 3D cloud radiative effects in AOD pixel retrievals from MODIS (Reid
et al., 2022) can be identified when comparing the expected natural variation as measured by 4STAR's predecessor (AATS-14; NASA Ames Airborne Tracking Sunphotometer) to the higher spatial variation retrieved by MODIS for the same matched scenes (Redemann et al., 2006). Here we investigate the common hypothesis that intensive aerosol properties are more homogeneous over larger length-scales than their extensive counterpart, as exemplified by a more horizontally homogeneous size-dependence, PM2.5, and its chemical composition than AOD in the southeast U.S.
(Kaku et al., 2018). We do this for aerosol measurements over the Korean Peninsula and its surrounding seas, where the aerosol type is dominantly mixed (urban-industrial, maritime, continental, sub-continental, biomass burning, and even some dust; Lee et al., 2018) due to multiple aerosol sources and transformation processes, contrary to other studies. This study has the potential to provide insight in air quality prediction, monitoring, and ultimately control.

The KORea-US Air Quality (KORUS-AQ; Crawford et al., 2021; Choi et al., 2019; Peterson et al., 2019) field study
measured atmospheric composition over the Korean peninsula and surrounding waters from May to mid-June 2016. During KORUS-AQ, the atmosphere was sampled by multiple airborne and ground-based remote sensing and in-situ measurements. Central to this paper is the airborne sunphotometer, 4STAR (Spectrometers for Sky-Scanning Sun-Tracking Atmospheric Research; Dunagan et al., 2013). 4STAR's main measurement is the spectral AOD representing the column above the aircraft. The following section summarizes the KORUS-AQ campaign, the 4STAR instrument,
the space-borne sensor, and the model used in this study. The methodology of defining length-scale consistency among different properties is presented in section 3, major findings and discussion in section 4, and conclusion in section 5. The appendixes provide some 4STAR measurement corrections, and comparison of AOD measurements during KORUS-AQ between 4STAR, GOCI, and MERRA-2.

## 2 Data Sources and Instruments

**2.1 Korea - United States Air Quality experiment (KORUS-AQ)**

As a result of the dramatic increase in economic and energy production in East Asia in the preceding decades, there has been a significant increase in fine particle and ozone pollution emission (Crawford et al., 2021; Zeng et al., 2019; Kim et al., 2020). The air quality has been impacted in the recent decade, particularly in the Seoul metropolitan area, with a population of ~25 million. A better understanding of the sources and evolution of the aerosol particles in the
region motivated a large-scale measurement campaign. The KORUS-AQ field study (Crawford et al., 2021), from May to June 2016, oversaw the deployment of the NASA DC-8 airborne research laboratory. KORUS-AQ was a joint effort by National Institute of Environmental Research of South Korea and NASA, and oversaw the deployment of

three research aircraft, extensive ground-based networks and three ships to observe and quantify the air quality in south Korea due to local and transported sources. The NASA DC-8 flew a total of 20 research flights over the South
Korean peninsula and surrounding waters (see Fig. 1). Portions of each research flight were conducted while keeping a near constant altitude (level legs) to quantify both the emission sources and the upwind/downwind variability of aerosol properties. These level legs that occurred mostly in the common corridors for air traffic and above research sites and common sources, enabled us to bound the spatial variability of aerosol properties (AOD, AE, and FMF) and thus giving a restriction of the combined aerosol microphysical processes and aerosol sources and sinks. These level
legs are illustrated in Fig. 1, and were identified through a running standard deviation of less than 5 meters in altitude for each 20 second segment.

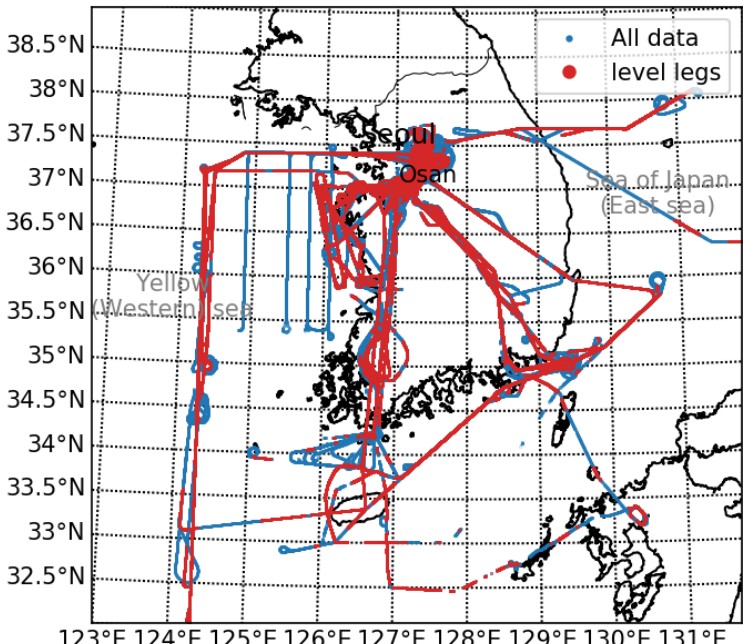

**Figure 1 - Map of NASA DC-8 flight paths during KORUS-AQ (blue), with the level (horizontal) legs with high-quality and cloud-free 4STAR data (red).**

During KORUS-AQ, the atmospheric conditions were divided into four main periods presented in Table 1 (summarized from Peterson et al., 2019).

|   | Date range | Short name | Description |
|---|------------|------------|-------------|
| **1** | 01–16 May | dynamic | dynamic meteorology and complex aerosol vertical profiles (7 flights) |
| **2** | 17–22 May | stagnation | stagnation under a persistent anticyclone (4 flights) |
| **3** | 25–31 May | extreme pollution | dynamic meteorology, low-level transport, and haze development with extreme pollution (3 flights) |
| **4** | 01–07 June | blocking | blocking pattern with a high-pressure ridge precluding any significant changes in synoptic meteorology (3 flights) |

**Table 1 - Description and time ranges of the four main meteorological periods during KORUS-AQ.**

Under each of these four time periods (dynamic, stagnation, extreme pollution, and blocking), the atmospheric dynamics and weather patterns have either promoted local production of aerosols (haze development in period 2 and 4, and mostly local sources in period 3) (Jordan et al., 2020) or brought in some long range transport of aerosols (dust transport in period 1, low-level pollution transport in period 2) from neighboring land and sea areas such as the Gobi desert, Shanghai, and Beijing (Peterson et al., 2019; Choi et al., 2019).

**2.2 Spectrometers for Sky-Scanning Sun-Tracking Atmospheric Research (4STAR)**

4STAR combines airborne sun tracking and sky scanning with diffraction spectroscopy and was integrated in the NASA DC-8 in the zenith port. This airborne sun tracker and sky radiometer has multiple operating modes, which are selected by the operator depending on the sky conditions above the aircraft. 4STAR observes spectral AOD (see LeBlanc et al., 2020), aerosol properties (e.g., single scattering albedo, scattering phase function, aerosol size distribution, aerosol refractive index) derived from skylight measurements following the AERONET retrievals (see Pistone et al., 2019), column trace gas density (e.g., water vapor, $O_3$, $NO_2$) using spectroscopic methods (see Segal-Rosenheimer et al., 2014), or cloud properties (cloud optical depth, effective radius, and thermodynamic phase) from transmitted light (LeBlanc et al., 2015, Smith et al., 2017). Here we focus on 4STAR's measurements of direct solar irradiance, AOD and derived products using its sun-tracking operating mode. 4STAR incorporates a modular sun-tracking/sky-scanning optical head with fiber optic signal transmission to rack mounted spectrometers with spectral resolution ranges from 0.5-1 nm from 350 nm to 1000 nm and 3-6 nm from 1000 nm to 1750 nm. The uncertainty of the AOD during KORUS-AQ is dependent on wavelength, solar angles, and varying corrections, but averages to 0.032 at 501 nm (0.033 at 452 nm to 0.027 at 1627 nm). This uncertainty includes corrections due to the deposition of material on the instrument window, fiber-optic rotating joint, gas phase absorption, and spectrometer non-linear correction, averaging to an uncertainty in AOD at 501 nm of 0.014, 0.0018, 0.0006, and 0.0005 respectively (described in appendix A), and the variability observed during the calibrations at Mauna Loa Observatory (0.67% standard deviation derived from six Langley-extrapolations, similarly method to Shinozuka et al., 2013 and LeBlanc et al., 2020). Recent advances in the fiber optic light path on 4STAR is also described in appendix A, which resulted in higher throughput and consistency in between calibrations (increasing number of viable Langley calibration from two (during TCAP; Shinozuka et al., 2013) to six) and removed the need for a temperature correction. Processing procedures and codes of the 4STAR raw measurements of solar direct beam into quality-assured AOD are presented by the 4STAR Team et al. (2020).

**2.3 Satellite-based remote sensing of AOD and Fine Mode Fraction (GOCI)**

The Geostationary Ocean Color Imager (GOCI), Yonsei aerosol retrieval (YAER) version 2 (Choi et al., 2018) provides a geostationary view of aerosol evolution over the Korean peninsula. This algorithm uses the reflectances measured by GOCI at eight spectral channels and retrieves hourly AOD at 550 nm and FMF (Fine mode fraction) over ocean and land. The native 0.5 km × 0.5 km resolution of GOCI is masked for clouds, inland waters, and highly turbid waters, and is aggregated to an AOD product with 6 km × 6 km resolution, by using the mean AOD from three aerosol models among total 27 models having best fits between calculated and measured spectral top-of-atmosphere reflectance. The GOCI YAER retrieved AOD is matched to the closest 4STAR observation in time and space along the DC-8 flight path during KORUS-AQ. The co-location criteria is a maximum of 30 minutes between the satellite and airborne observations, and a maximum distance of 3 km. The expected deviation of AOD during this 30 minute lag is lower than 0.06, as identified by the variogram analysis of ground-based sunphotometers (Sayer, 2020), and confirmed in Korea at a few AERONET stations (Park et al., 2020b). Detailed GOCI aerosol retrieval algorithm, AOD features, and evaluation during the campaign are described in Choi et al. (2019). GOCI YAER version 2 has been compared to AERONET measurements over 5 years (Choi et al., 2018), reporting a root mean square error (RMSE) of 0.16 and R=0.91 for AOD over land with N=45 643, and slightly lower (RMSE=0.11, R=0.89) over ocean neighboring AERONET sites (N=18 499). For FMF, the GOCI YAERv2 retrievals are slightly biases towards more coarse mode particles, with comparisons to AERONET for AOD>0.3 having R=0.623 over land.

**2.4 MERRA-2 aerosol modelling and computation of AOD**

The emission, evolution, transport, and removal of aerosols are represented by a reanalysis system, NASA's Modern-Era Retrospective Analysis for Research and Applications, version 2 (MERRA-2; Buchard et al., 2017; Randles et al., 2017). For understanding the vertically integrated AOD, and AE intensive properties, we used the hourly resolved aerosol diagnostics with a spatial resolution of $0.5\,° \times 0.625\,°$ (roughly 59 km between two pixels) (GMAO 2015). In MERRA-2, aerosols are simulated by GEOS model driven by assimilated meteorology fields and assimilates bias-corrected AOD derived from AVHRR and MODIS radiances and AOD from MISR and AErosol RObotic NETwork (AERONET; Holben et al., 1998) to build a four-dimensional gridded model representation of the real world's sparse observations.

The reanalysis AOD and AE values are interpolated in time and space (4D) along the DC-8 flight trajectory from samples of a global analysis with a spatial resolution of 0.3125-degree longitude by 0.25-degree latitude at a temporal resolution of 3 hour (Collow, et al., 2020). This gives a maximum temporal difference of 90 minutes and a maximum distance difference of approximately 12 km between airborne observations and reanalysis grid point. MERRA-2's total AOD include five species of aerosol: dust, sea salt, organic carbon, black carbon, and sulfates. The size distributions of sulfates, organic, and black carbon aerosols are all modeled as lognormal distributions. The mode of the lognormal distribution for the dry aerosol is 0.0695 μm, 0.0212 μm, and 0.0188 μm for sulfate, organic carbon, and black carbon, respectively. The minimum and maximum dry particle radii ($R$) are 0.005 μm and 0.3 μm. The size distributions for sulfate and the hydrophilic portions of organic and black carbon change with relative humidity

according to a growth factor (GF) taken from the Global Aerosol Data Set (GADS; Köpke et al. 1997). The computation of the aerosol size within this model is achieved through combining the dry aerosol size with the GF. For some RH equal to $n$ mode of the lognormal distribution ($R_M^n$) and the maximum radius ($R_{Max}^n$) is given by:

$$R_M^n = GF^n R_M^{dry} \tag{1}$$

$$R_{Max}^n = GF^n R_{Max}^{dry} \tag{2}$$

The dusts and sea salt aerosols have their size defined through a five-bin system, ranging from 0.1 μm to 10 μm for dust, and 0.03 μm to 10 μm for sea salt. Dust is modeled as hydrophobic, while sea salt is hydrophilic. The particle size distribution in each sea salt bin is described by equation 2 in Gong (2003). Particle growth as function of RH is from Gong et al. (1997) (see equation 3 and Table 2). The size of each sea salt bin changes as the particles grows.

While not strictly super-micron aerosol sizes, majority of sea salt and dust are considered here as corresponding to the optically derived coarse mode aerosols, and the sulfates, organic, and black carbon aerosol are considered as part of the optically defined fine mode aerosol. The modeled FMF is taken to be the fraction of total AOD that comes from sulfate, organic carbon, and black carbon.

**2.5 In situ airborne aerosol extinction measurements from LARGE (NASA Langley Aerosol Research Group**
**Experiment)**

In situ optical aerosol measurements from the NASA DC-8 were obtained by LARGE (NASA Langley Aerosol Research Group Experiment; Ziemba et al., 2013) using a combination of TSI-3563 nephelometers for scattering coefficients, Radiance Research 3-wavelength PSAP (Particle Soot Absorption Photometer) for absorption coefficients (with wavelength-dependent corrections from Virkkula, 2010), and the impact on scattering by
230 hygroscopic growth using nephelometers measuring aerosols at dry (<20% RH) and humid (80% RH) conditions. Scattering coefficients were measured at 450, 550, and 700nm and absorption coefficients at 470, 532, and 660nm. The aerosols are brought into the aircraft for observation via a shrouded solid diffuser inlet which has a cutoff at 5 μm dry aerodynamic diameter (McNaughton et al., 2007). The total ambient aerosol scattering coefficient and AE is calculated by correcting dry scattering measurements to ambient relative humidity using a gamma relationship and
235 measured hygroscopicity (Ziemba et al., 2013). Total ambient extinction coefficients are the sum of ambient scattering and dry absorption coefficients. These measurements are sampled at 1 Hz, with an effective distance between sample points of roughly 130 m, and reported at standard temperature and pressure (273K and 1013mb) for quantifying the variability over spatial and temporal domain, even at different altitudes. We use here the total extinction coefficient at 532 nm and the AE calculated from 550 nm to 700 nm.

These in situ measurements of the aerosol optical properties represent the aerosol environment at the DC-8 flight altitude, unlike the column observations by the 4STAR and GOCI, and as reported by MERRA-2. Additionally, the column measurements (AOD, AE, and FMF) are derived from optical remote sensing, which is sensitive to aerosol particles over a broad range of sizes at ambient conditions (Hou et al. 2020), while the in situ observations have lower sensitivity to particles larger than 5 μm in aerodynamic diameter, owing to the inlet efficiency.

## 3 Methodology

The AOD, AE and FMF observed and modelled during KORUS-AQ are compared to each other first by setting a common time base for comparison. Here we use the AOD observations aboard the NASA DC-8, which has the finest spatial and temporal resolution of all data sources used in this study. Each observation of AOD spectra by 4STAR along the flight path is matched to the nearest LARGE in situ observation, nearest satellite retrieval pixel (GOCI), and to the 4D interpolation of the model reanalysis (MERRA-2). Only the level legs are used here to build this collocated dataset (see Fig. 1). This combined collocated dataset is used to investigate the spatial distribution, spatial variation and representative nature of the AOD, AE and FMF during KORUS-AQ.

### 3.1 Autocorrelation distance

To identify the distance at which one measurement can be best represented by itself, or to what distance an aerosol property is consistent, we use the autocorrelation metric, as popularized by Anderson et al. (2003), and used by Redemann et al. (2006) and Shinozuka and Redemann (2011). The formulation from eq. 5 by Shinozuka and Redemann (2011) is used here (Eq. 3). Autocorrelation is the correlation coefficient among all data pairs $x_j$ and $x_{j+k}$ within a set that exists at a separation, or lag, of k. That is,

$$r = \frac{\sum_j^N[(x_j - m_{+k})(x_{j+k} - m_{-k})]}{(N-1)std_{+k}std_{-k}},$$  (3)

where k indicates the spatial lag (or distance), $m_{+k}$ and $std_{+k}$ denote the mean and standard deviation respectively, of all data points that are located a distance of +k away from another data point, and $m_{-k}$ and $std_{-k}$ are the corresponding quantities for data points located a distance of $-k$ away from another data point. Thus, one can reproduce the autocorrelation at various distances from each sample within a set, here understood as a flight leg.

To relate the autocorrelation distance of samples to the greater physical characteristics of the aerosol itself, a measure of the sampling bias can be estimated. This is done similarly to Shinozuka and Redemann (2011), where a Monte Carlo subsampling of the dataset is chosen, and the standard deviation of the autocorrelation distances is calculated. We used a randomly chosen portion (30%) of all legs (304 level legs spanning all 20 research flights) to calculate the autocorrelation distances, which were iterated 50 times to calculate the standard deviation at each distance interval. We also allow each discretized bin, or lag distance k, to be within 20% of the actual separation distance, effectively making the width of the bin 20%, thereby increasing the data set for calculation. As example, the lag distance k of 1.0 km encompasses all points within a segment that are separated by a distance from 0.8 km to 1.2 km. To remove influence of instrumental noise the shortest autocorrelation bins were ignored, representing a distance of 80 m, with a bin ranging from 64 to 96 m.

### 3.2    Spectral deconvolution of AOD for obtaining fine mode fraction and relation to Ångström Exponent

Inferring particle size information from spectral AOD has been widely demonstrated (O'Neill et al., 2003; O'Neill et al., 2008; Eck et al., 2010). Schuster et al. (2006) points out that for a bimodal distribution, AE may reflect particle size (especially the short wavelengths) or fine mode fraction (reflected more in AE at longer wavelengths). The larger

aerosol particles have a flatter spectral shape, and greater impact on extinction at longer wavelengths than small aerosol particles compared to a steeper spectral response in AOD for a similar mid-visible AOD. Generally, this

produces low AE (roughly below 1) when the aerosol optical depth is dominated by the larger aerosol particles, and conversely high AE (roughly greater than 1) for the column dominated by smaller particles. However, when considering changes in not only the size but also chemical composition, low AE (but still mostly above 1) has been observed for fine mode dominant aerosol that are humidified in the Korean Summer (Koo et al., 2021). Koo et al. (2021) describe one of the downfalls of using solely AE for quantifying the fraction of aerosol optical depth that is

dominated by fine particles (FMF), and also illustrates the impact of different wavelength ranges used in evaluating AE, notably caused by curvature in the spectral AOD (other examples of variance in AE depending on the range of wavelengths used are presented by LeBlanc et al., 2020 and by Eck et al., 1999). Curvature in spectral AOD can also be caused by the relative proportions of the fine and coarse mode aerosols or the FMF (Eck et al., 1999, Yoon et al., 2012). While AE does relate generally to aerosol size, it is a proxy for FMF. The AOD spectral curvature information,

with the assumption that coarse mode aerosol consistently result in low AE, is further exploited in the Spectral Deconvolution Algorithm (SDA; O'Neill, 2003; O'Neill et al., 2001a, 2001b), which is used in AERONET for retrieving FMF.

We apply SDA (O'Neill et al., 2008) on the 4STAR sunphotometer, which unlike AERONET, samples the AOD using spectrometers and allows for wavelength choice. Here the fine mode fraction is still reported at 500 nm, as previously

evaluated (e.g., Eck et al., 2010), but we expand the input wavelengths of the AOD spectra to include AOD at 452, 501, 520, 532, 550, 606, 620, 675, 781, 865, 1020, 1040, and 1064 nm, unlike the five typically used from AERONET (O'Neill et al., 2008). Although we used many more wavelengths to characterize the spectral derivative, the shortest wavelength is omitted but is expected to produce results similar to the standard AERONET wavelength set with RMS differences in retrieved fine mode AOD of less than 0.01 (O'Neill et

al., 2008). While the SDA does not directly evaluate the volumetric fine and coarse mode of aerosol size distribution, this optical equivalent is nearly linearly proportional to the volumetric sizes of the aerosol distribution (e.g., Hou et al., 2020). However, the SDA only assumes a bimodal distribution (fine and coarse), there may be instances of a middle-mode aerosol size due to cloud or fog processing (e.g., Eck et al., 2020) that lie on the boundary between the two modes assumed in SDA. The AE in Korea is also directly

proportional to the FMF except for fine mode aerosol in environments of high relative humidity (above 80%) (Koo et al., 2021).

To quantify the uncertainty in AE ($\sigma_{AE}$) and FMF ($\sigma_{FMF}$), we used a propagation of the measured AOD probability distribution. For each measurement of AOD at time ($t$) and wavelength ($\lambda$), a probability distribution ($P_{AOD}(t,\lambda)$) is built from the measured AOD($t$, $\lambda$), a reference probability distribution ($P_{ref}(\lambda)$), the uncertainty in AOD

measurement ($\sigma_{AOD}(t,\lambda)$), and the standard deviation of the reference probability distribution ($std_{ref}(\lambda)$). Where,

$$P_{AOD}(t,\lambda) = AOD(t,\lambda) + \left(P_{ref}(\lambda) - \overline{P_{ref}(\lambda)}\right) \times \sigma_{AOD}(t,\lambda)\Big/ std_{ref}(\lambda) \; , \qquad (4)$$

represents the linear transformation of the reference probability distribution and its mean $(\overline{P_{ref}(\lambda)})$ to the AOD measurement probability distribution, such that the mean of $P_{AOD}(t,\lambda)$ is equal to the AOD value, and its standard deviation is equal to the measurement uncertainty, while keeping the relative relationship of the probability distribution for each wavelength. The $P_{ref}(\lambda)$ is built from 1023 points subset of the high-altitude level legs (greater than 6 km), low AOD (<0.07 at 500 nm), AOD spectra measured by 4STAR. This represents the instrument's measuring variability, while conserving the high spectral covariance afforded by the spectrometer design of 4STAR. From this, a probability distribution of AE and FMF ($P_{AE}(t)$ and $P_{FMF}(t)$) are computed by passing each individual AOD spectra contained within the distribution ($P_{AOD}(t,\lambda)$) to the AE function ($f_{AE}(AOD(\lambda))$ and FMF function ($f_{FMF}(AOD(\lambda))$), following

$$P_{AE}(t) = f_{AE}(AOD(t,\lambda)) \; for \; AOD(t,\lambda) \; in \; P_{AOD}(t,\lambda) \; , \text{and} \qquad (5)$$

$$P_{FMF}(t) = f_{FMF}(AOD(t,\lambda)) \; for \; AOD(t,\lambda) \; in \; P_{AOD}(t,\lambda) \; . \qquad (6)$$

The AE function is defined by the linear fit of the log of AOD over the log of the wavelengths, while the FMF function is defined by the SDA. Finally, the uncertainty in AE ($\sigma_{AE}$) and FMF ($\sigma_{FMF}$), are computed by the standard deviation of the probability distribution ($P_{AE}(t)$ and $P_{FMF}(t)$) for each time step, thus representing the expected uncertainty of these computed values given 4STAR's uncertainty due to instrument variability and accuracy.

## 4 Results and Discussion

Aerosol distribution over and surrounding the Korean peninsula has contributions and influences from a multitude of varying sources. Recent publications show that the retrieved black and brown carbon aerosol from polarized satellite retrievals over the Korean land mass has distinctly different mass concentration and ratio of black to brown carbon in March through May compared to June through August (Li et al., 2020). Although black and brown carbon aerosol are not the only aerosol types found in this region, their mass concentrations are reported to vary by a factor of 5 over Korea, forming multiple gradients, regional maxima and mimina within the boundaries of the peninsula (Li et al., 2020). This type of spatial variance is indicative of multiple processes impacting aerosol in the region. An aerosol process largely impacting the region is the high relative humidity in the summer, which increases the size of the small aerosol particles stemming from pollution and secondary organic formation (Koo et al., 2021).

### 4.1 Overview of AOD variability during KORUS-AQ

4STAR measured an average AOD at 501 nm of 0.36 with a standard deviation of 0.12, and an average AE of 1.11 with a standard deviation of 0.15 during KORUS-AQ, when flying below 1000 meters.

Aerosol spatial distribution during KORUS-AQ is quantified by spatially binning (0.44° latitude and 0.33° longitude) the AOD observations for the entire period and obtaining the mean and standard deviation of each binned AOD (see

Fig. 2). The largest AOD averages observed by 4STAR (below 1000 m altitude) occurred near Seoul, and over the Yellow Sea, on the western coast of Korea. Lower averages were found along the southern coast. The mean AOD at 501 nm over Seoul during KORUS-AQ is between 0.29 and 0.45, from the northernmost spatial bin to more southern

bins encompassed within Seoul metropolitan area. This range is similar, albeit a bit low compared to the climatologically average observed values from a ground site for May and June for the decade 2007 – 2017 (~0.45 and 0.36 respectively; Choi and Ghim, 2021). The regions with the largest variability in AOD, as determined by the standard deviation in each of the binned statistics, are observed on the Western Sea, and near Seoul. Directly south of Seoul, where there are large industrial regions, a larger than average standard deviation is also observed. The number

of samples aggregated within each spatial bin to build these statistics are represented by the size of the square symbols in Fig. 2 right panels, while the number of days sampled are represented by the size of the circle symbols. The greater Seoul region shows a high number of observations, followed by the sea west of Korea, along the western coast, and over land on the corridor between Seoul and Busan (southeast Korea). Although the remaining regions show some variability, the number of samples are lower than 1000 (roughly 3 days), with the lowest sampled region (by number

of samples or days sampled) coinciding with the smallest average AOD. The standard deviation is only weakly correlated to the number of samples within a bin at a Pearson correlation coefficient of $R^2 = 0.13$, with a higher standard deviation for larger number of samples where every additional 403.2±99.9 samples, or 0.9±0.3 days sampled, there is an additional 0.1 standard deviation in AOD. The AE is more dependent on the number of samples or days sampled, with an $R^2 = 0.19$ and $R^2 = 0.36$ respectively, with an increase in standard deviation of AE by 0.1 for locations

per 550±111 number of samples, or 2.0±0.3 days sampled. However, this relationship does not seem to hold with the matched MERRA-2 samples, where a higher number of samples or days sampled does not directly translate to higher standard deviation.

The AOD spatial trend which is higher in the northwest, and lower along the southern coast is reproduced in the MERRA-2 reanalysis, for the collocated pixels in time (within three hours) and space to the observations. However,

MERRA-2 does not seem to reproduce the same high average AOD over Seoul and the industrial region to the south and tends to overpredict the AOD along the southern coast. Similarly, the geostationary observation by GOCI, show a similar trend of high AOD over the Western Sea and Seoul, with lower values on the Eastern Sea, and south, but the number of high-quality retrievals that are co-located with the NASA DC-8, while flying below 1000 m, are low.

KORUS-AQ Aerosol properties spatial distribution

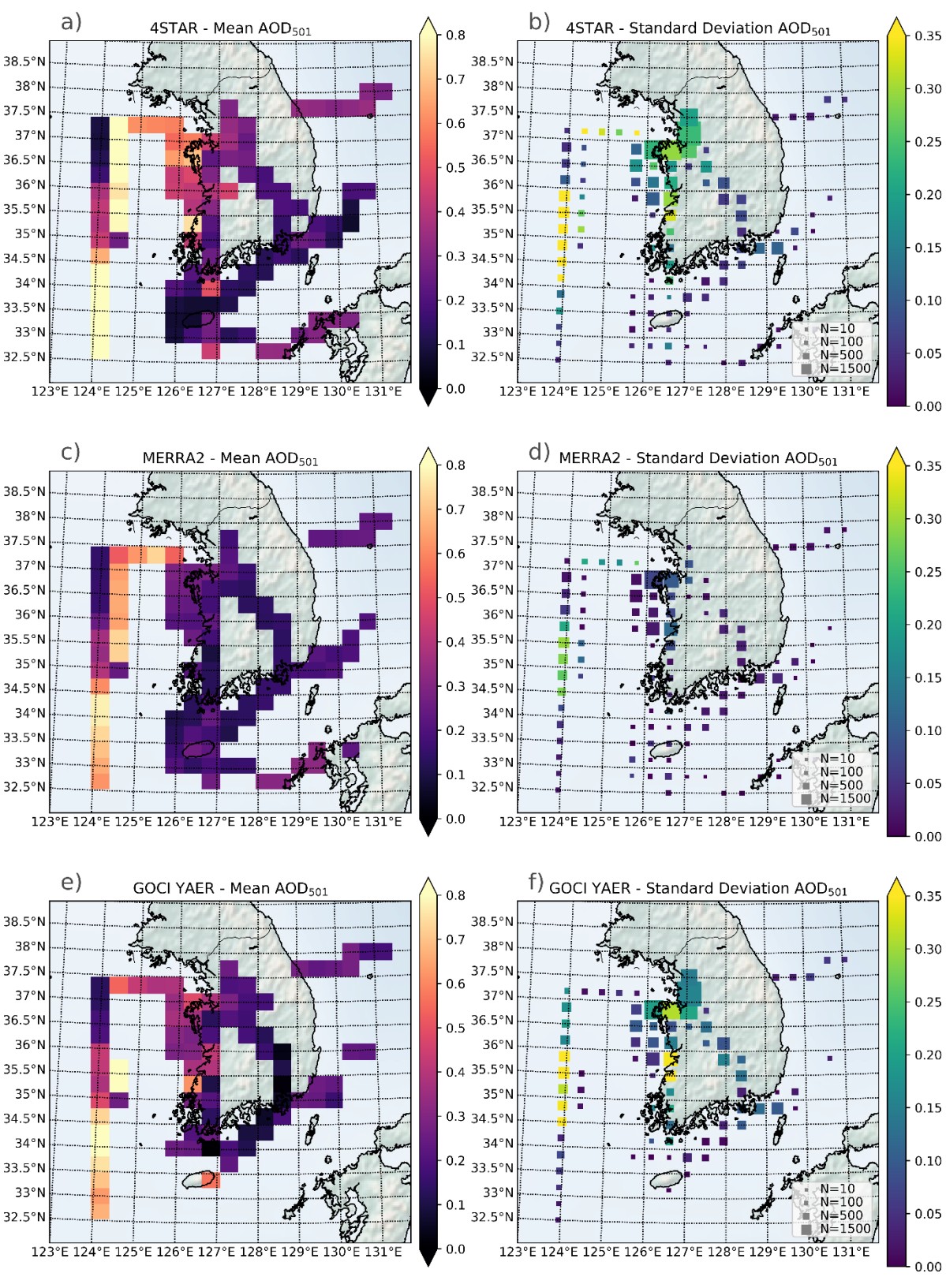

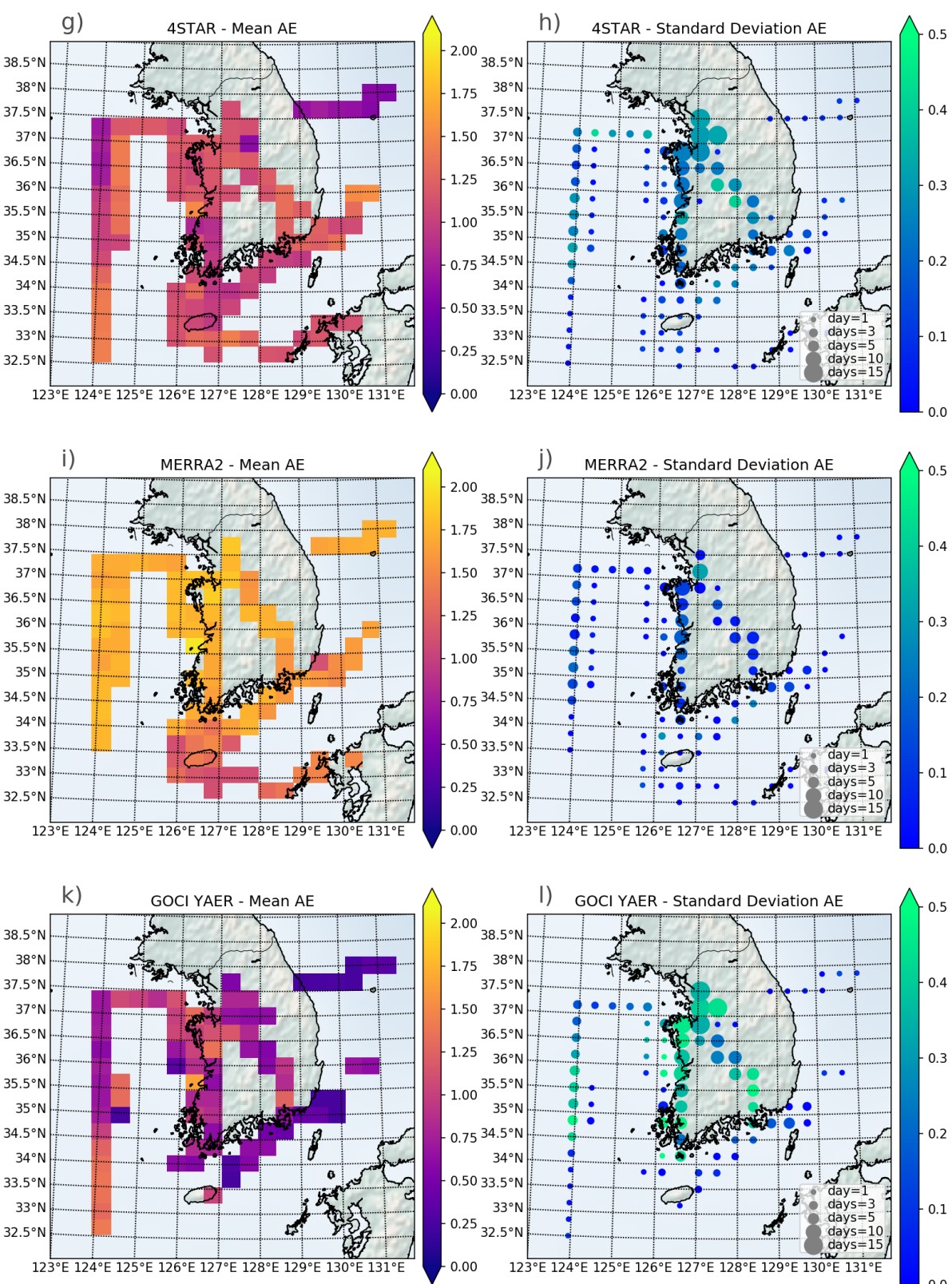


**Figure 2 – The spatial distribution binned by 0.44° latitude and 0.33° longitude for AOD measured by 4STAR (a-b), MERRA-2 (c-d), and GOCI (e-f) during KORUS-AQ matched to when the NASA DC-8 flew below 1 km altitude. The**

**average AOD in each spatial bin is on the left panels (a, c, and e), while the right panels (b, d, and f) showcase the standard deviation of the observations within each spatial bin. The number of samples is represented by the size of the square symbol**

**for a, b, c, d, e, and f, while the number of days sampled are represented by the size of the circle for g, h, i, j, k, and l. Similarly for AE in spatial bins (g, i, and k), and its standard deviation (h, j, and l)**

The aerosol properties during KORUS-AQ not only varied based on location, but also based on the meteorological period, see Table 1. The AOD was highest during the extreme pollution meteorological period and lowest during the stagnation period (Fig. 3). The dynamic/transport period has a dual peak of AOD, with a main peak below 0.1, and

secondary peak near 0.3, all while having the flattest average AOD spectra which is linked to a high fraction of coarse mode aerosols (consequently lowest AE in the legend of Fig 3b). The secondary peak of AOD measured during the dynamic period, for AOD higher than 0.2, has an average AE of 0.57 likely indicative of presence of coarse mode aerosol like dust influencing the highest AODs. The blocking period, while not the highest average AOD, had a mode at 0.5, and had the steepest slope with respect to wavelengths, indicating a dominance of the fine mode aerosol in the

column (Fig. 3b). The AOD in Fig. 3b showcase the spectral dependence over the various periods, particularly with respect to their slope as evaluated over the entire range reported here, which minimize the impact of AOD variations at any one wavelength. The AE of different meteorological periods follows a similar tendency when measured by the ground-based AERONET in Korea during KORUS-AQ (Eck et al., 2020), for higher AODs with a greater proportion of fine mode aerosols going from May into June (Dynamic → Stagnation → Extreme pollution → Blocking,

meteorological periods in order of May to June).

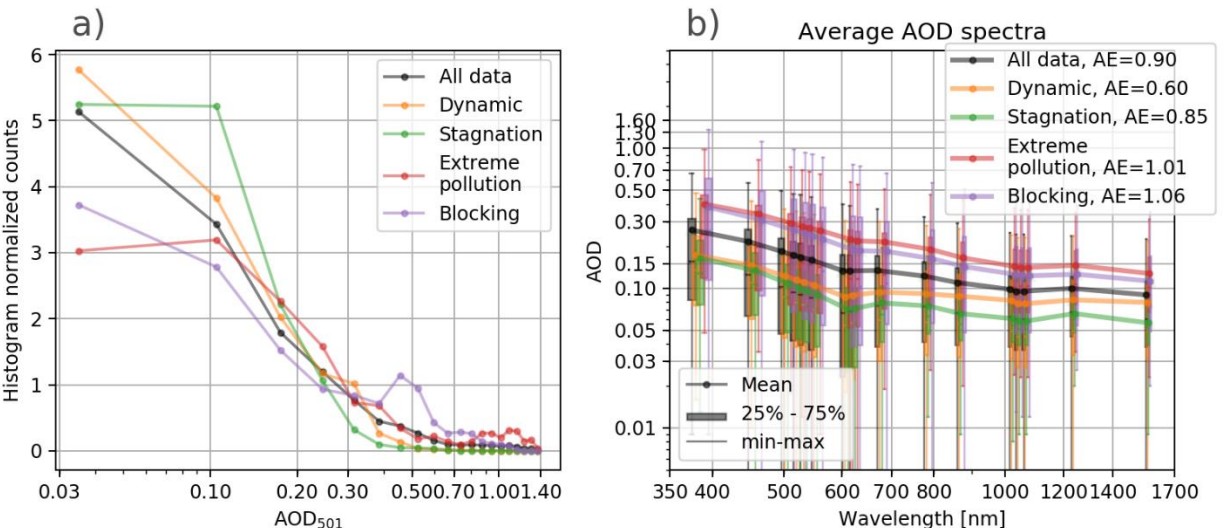

**Figure 3 - (a) Histogram of AOD at 501 nm measured by 4STAR distribution from KORUS-AQ, separated by meteorological periods. (b) Corresponding AOD average spectra for each meteorological period, with the error bars denoting the range of AOD (excluding outliers) during that time period, with the thicker bars denoting the interquartile**

**range. The square symbols and error bars are slightly shifted from each other for clarity. The AE in b) is calculated from the average spectra of each respective meteorological period from 453 nm to 870 nm.**

The uncertainty in AOD also varied with enhancements of the uncertainty during the periods of window deposition (see appendix A.1.3) that are also related to the larger AOD (see Fig. 4a). For most of the measurements, the mean and median uncertainty in AOD is near 0.03, except for observations with AOD near 1.2, and 0.3, which have higher

mean than median uncertainty. The uncertainty in AE and FMF are calculated from all the AOD measurements using a propagation of the probability distribution, described in section 3.2. The AE and FMF uncertainty are both inversely

proportional to the measured AOD (particularly for their medians, see Fig. 4b and 4c), peaking when AOD at 501 nm is below 0.025 with lower average AE and FMF than for larger AOD bins. At the mean measured AOD of 0.36, we find the median uncertainty of AOD, AE, and FMF are 0.028, 0.106, and 0.057 respectively. Median uncertainty in

AE of 0.15 occurs when AOD is greater than 0.2 (which corresponds to roughly 63% of the samples below 1000 m), and is associated with a median uncertainty in FMF of 0.07.

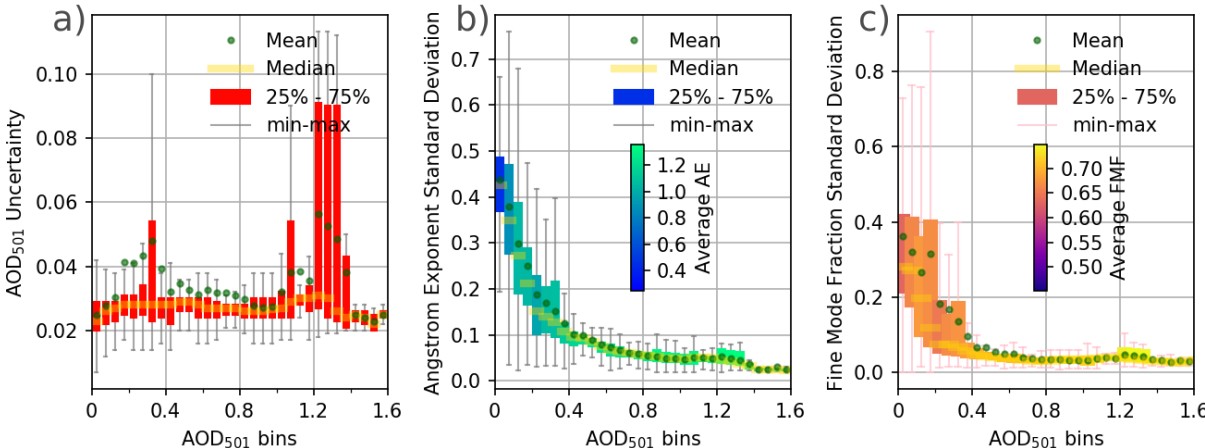

**Figure 4 – 4STAR measurement uncertainty of (a) AOD at 501 nm, (b) AE, and (c) FMF binned by measured AOD at 501 nm for all quality-assured observations during KORUS-AQ. The AE and FMF uncertainty are computed error propagation**
**by using a measurement probability distribution function. The shading of the color in (b) and (c) represent average value of AE and FMF in that AOD bin.**

**4.2 Along flight-path 4STAR measurements matches regional averages from GOCI and MERRA-2**

During KORUS-AQ, the NASA DC-8 was deployed for 44 consecutive days, of which 20 days were sampling days. While the NASA DC-8 is heavily instrumented to accurately observe the atmospheric composition, it was unable to
measure the entire region during each flight. It is in consequence unclear how representative these airborne samples were of the broader region. To answer this question, we compared the overall average and the space and time matched observations observed by GOCI or modeled by MERRA-2 to the valid samples by 4STAR on board the NASA DC-8 (Fig. 5). The regional averages are defined by daily averaging of the GOCI or MERRA-2 over all of South Korea, while the flight averages are only the daily averages for the aerosol properties from GOCI and MERRA-2 that are
collocated to the NASA DC-8 flight path for the days that were sampled by the airborne platform. While the flight averages miss the sporadic aerosol events apparent in the regional average time series from both GOCI and MERRA-2, the overall trend in AOD over the field study time period is well represented by the flight averages. The difference between the GOCI flight average and the GOCI regional average showcases how well representative the flight sampling is to the broader regional average over Korea and neighboring waters.

Differences between averages from 4STAR and from the GOCI flight averages illustrate potential differences between the GOCI retrievals and the 4STAR measurements. Notably, the mean and median average AOD from GOCI regional and flight averages are nearly identical, while 4STAR AOD averages have a lower mean and median by up to 0.043. Inversely the FMF from 4STAR is higher than GOCI by up to 0.07, while the regional and flight averages are within 0.02 of each other. The low bias of GOCI for FMF and AE as compared to 4STAR is expected (see Fig. B1),

particularly when the AOD is low (scene analysis e.g., Fig. 7 of Choi et al., 2016), since GOCI YAER retrieval preferentially selects coarse-mode dominant aerosol models when there is limited signal. This low bias from GOCI FMF and AE is also observed when comparing to AERONET retrievals (Choi et al., 2018). In addition to the already known AERONET comparisons over land, we find that the coarse mode AOD from GOCI has a lower RMSE over ocean (RMSE=0.093) than land (RMSE=0.112), albeit with relatively low correlation ($R^2$ of 0.058 and 0.066

respectively). This low correlation is accompanied by nearly flat slope when comparing GOCI to 4STAR coarse mode AOD over ocean (0.26±0.05), and less so over land (0.49±0.12), as estimated using a bivariate linear fit (York et al. 2004). The fine mode AOD is much closer to the expected 1:1 line with slopes of 0.83±0.03 and 0.78±0.09 over ocean and land respectively, and low biases of 0.05 and 0.06 for fine mode AOD (see Fig. B2). Albeit having a difference between 4STAR and GOCI, the small difference between GOCI regional and flight averages for FMF reinforces the

representativeness of the 4STAR samples within the Korean region.

The MERRA-2 mean regional AOD for the entire period of KORUS-AQ is biased high as compared to 4STAR AOD by nearly 0.1, but when matched to flight days MERRA-2 is biased low on average by 0.08. The regional average MERRA-2 AOD is likely overestimated due to higher AOD on the later period of KORUS-AQ, with less frequent flights (see Fig. 5e). Similar high bias is observed when comparing the MODIS dark-target AOD retrievals, which is

assimilated by MERRA-2, to AERONET measurements during KORUS-AQ (Choi et al., 2019). Both the regional average and the subsampled flight days AE from MERRA-2 are overestimated as compared to 4STAR by nearly 0.25, suggesting a higher representation of fine mode aerosol in MERRA-2 than observed by 4STAR, while the opposite occurs for GOCI. A more thorough comparison of the AOD between 4STAR, GOCI, and MERRA-2 is presented in appendix B.

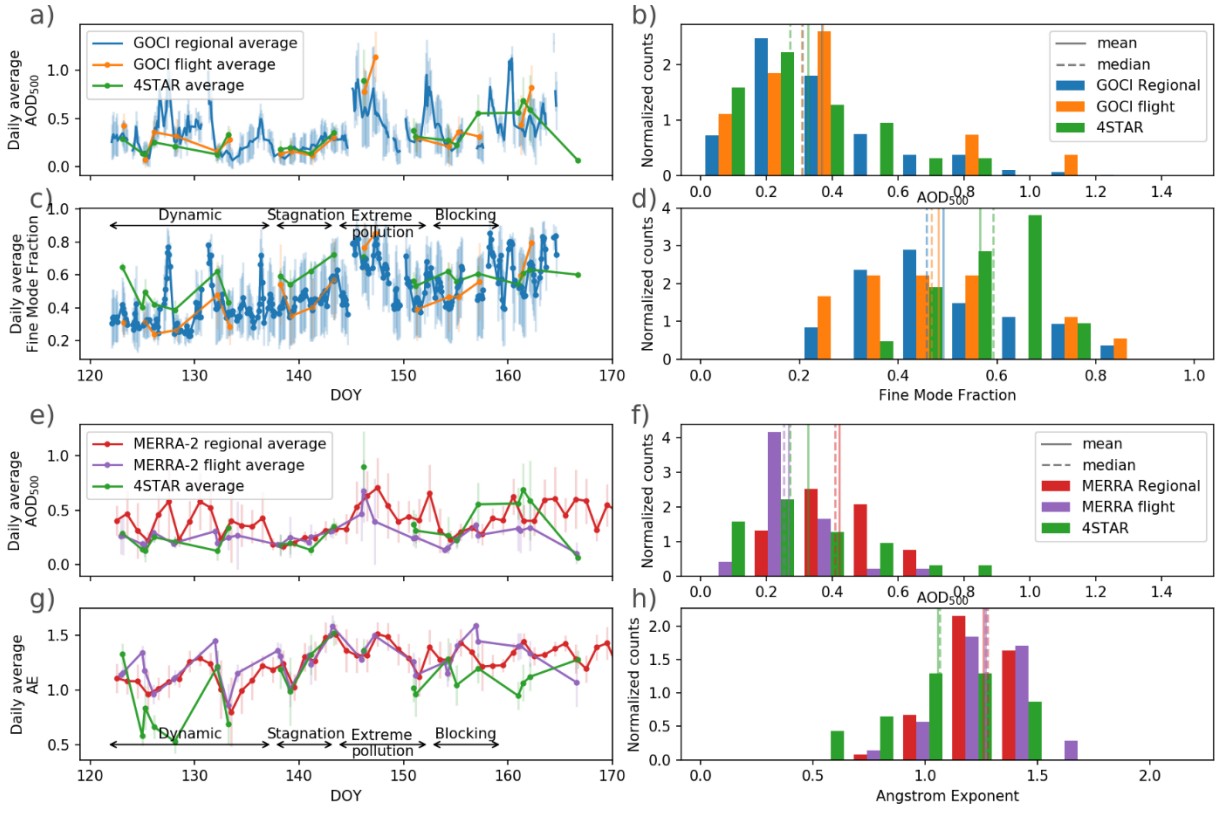


**Figure 5 - The average AOD at 500 nm and the FMF or AE for the region observed during KORUS-AQ by 4STAR, GOCI (a,c), and MERRA-2 (e,g), as a function of day of year (DOY). The continuous time trace of the AOD (a) and FMF (c) for the GOCI regional average (for spatial bins centered from 33.8°N to 37.6°N and 124.3°E to 129.4°E) is compared to the subset of GOCI matching in space and time (GOCI flight average) with the 4STAR for each day. 4STAR measurements**
**are also compared to the daily averaged time trace for MERRA-2 AOD (e) and MERRA-2 AE (g), for either regional averages or the MERRA-2 subset for flight averages. Histogram of the averages based on regional and flight for GOCI AOD (b), GOCI FMF(d), MERRA-2 AOD (f), and MERRA-2 AE (h) presented in the time trace, with vertical solid lines denoting the mean, and dashed lines - the median. The meteorological periods are identified in (c,g) by the span of arrows relating to the edges DOY for each period.**

During the 20 flight days, only two flight days have a difference of average AOD between the regional average and

the flight average of GOCI data greater than the regional standard deviation (see Fig. 6). While the regional standard

deviation can be sometimes large (0.3 around DOY 145, grey shaded region in Fig. 6), the average is closer to 0.1

during KORUS-AQ. There is no clear preference between over and underestimation of the GOCI flight average subset

compared to the regional average, similar to the difference between 4STAR observations and GOCI flight average

subset. Thus we can say that the sampling of AOD during the flights are representative of the greater Korean region

for 18 out of 20 days.

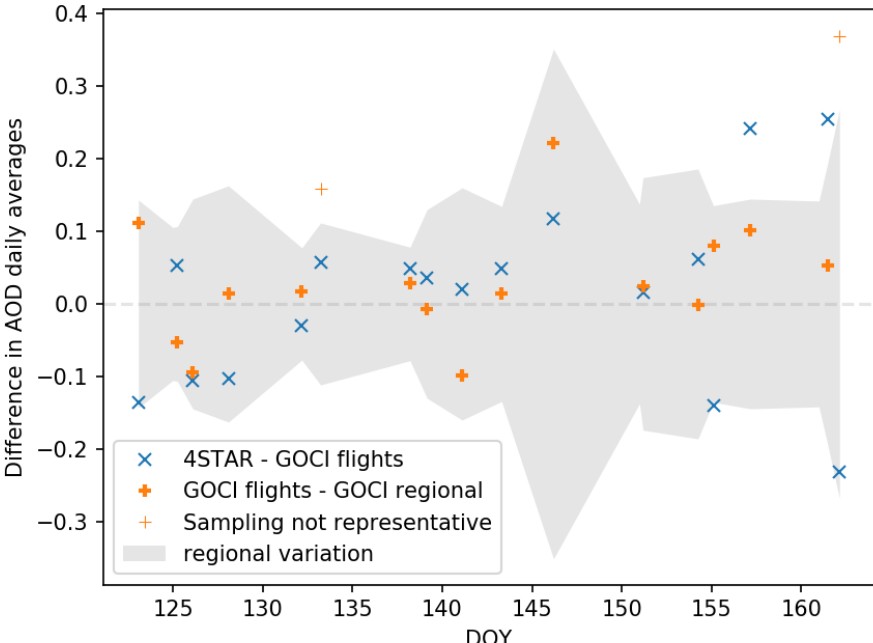

**Figure 6 - Difference in daily average AOD as a function of the DOY depending on the sampling: GOCI regional averages, GOCI flights - subset to match spatial and time of flight observations, and differences between 4STAR matched to the GOCI flights. The grey shaded regions indicate one standard deviation of the AOD in the region as observed by GOCI.**

**4.3 Vertical variations in aerosol distribution**

Aerosol from local sources and transported from sources further away are likely to stratify vertically and have different optical properties, depending on aerosol age and source (e.g., dust from Gobi Desert). During KORUS-AQ, we observed that the largest contributor to total column AOD is near the ground, below 500 m (see Fig. 7) and has the largest slope of AOD with respect to wavelength (see below 0.5 km line in Fig. 7a).

The averaged profiles presented in Fig. 7a match individual profiles from the frequent missed-approaches during each flight (3 times per flight near Seoul), the landing and take-off profiles at Osan, and low flight maneuvers over water, particularly for AOD below 500 m. In Fig. 7a and as reported by Choi et al. (2021) when comparing 4STAR to AERONET sites, the difference in AOD from surface to 500 m is roughly 0.1 at 500 nm, but with a highly consistent AE throughout that lower layer.

The lowest aerosols have the highest fraction of fine mode, as would be expected from pollution-based aerosol, and new aerosol formation. The vertical distribution of AOD is a column measurement, representing the aerosol content between the measurement altitude and the top of atmosphere, thus the lower AOD spectra also incorporates the influence of the elevated aerosol layers, which are mostly influenced by coarser aerosol, as identified through the much lower AOD spectral slope. This is consistent with the observed aerosol transport at low altitudes consisting of pollution from more directly west, while higher altitudes flow was dominated by transport from more northerly regions (Peterson et al., 2019). Natural stratification from the combined AOD statistics during KORUS-AQ, resulted in a slope change of the AOD spectra with observation altitude for AODs below 500 m, above around 2 km, and above 5 km. The distribution of altitudes at which each sample is measured and the altitudes where each level legs are similar (Fig. 7c), with some under-representation of level-leg to all samples taken between 3 and 7 km.

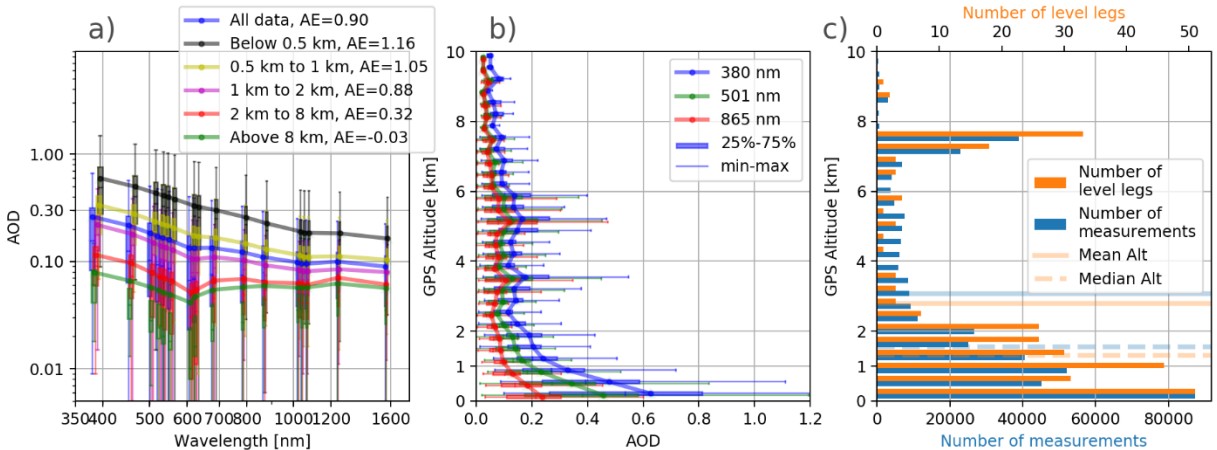

**Figure 7 - Aggregated AODs observed during KORUS-AQ as a function of observation altitude, for (a) with average AOD spectra, and (b) binned vertically for a subset of wavelengths. The range in binned values are presented by the error bars, while the thicker bar denotes interquartile range (25%-75%). The number of spectra per height bins in a) is 64 736, 41 821, 63 130, 121 569, and 31 076 from lowest to highest respectively. (c) The histogram of the altitude by number of data points (bottom axis) and by number of level legs (top axis), with the mean and median altitudes indicated by solid and dashed lines with the respective colors.**

The stratification of the AOD as a function of altitude can be further examined in terms of the aerosol's fine and coarse mode contribution to the total AOD, which is obtained through the SDA extraction of the statistically aggregated observations of the AOD (Fig. 8). The impact on the vertical location of the observation is apparent in the probability distribution of fine and coarse mode AOD, where the fine mode AOD is responsible for nearly all AODs greater than 0.6, and AODs greater than 0.6 are only observed below 500 m. This large fine mode fraction is absent in observations of the atmospheric column starting at 2 - 5 km up to top of the atmosphere. There is also a shift in the peak coarse mode fraction from ~0.15 to ~0.05 between 0.5 km and the 2-5 km layers (not shown), suggesting that the lowest portion of the atmosphere hosts a sizable portion of coarse mode AOD.

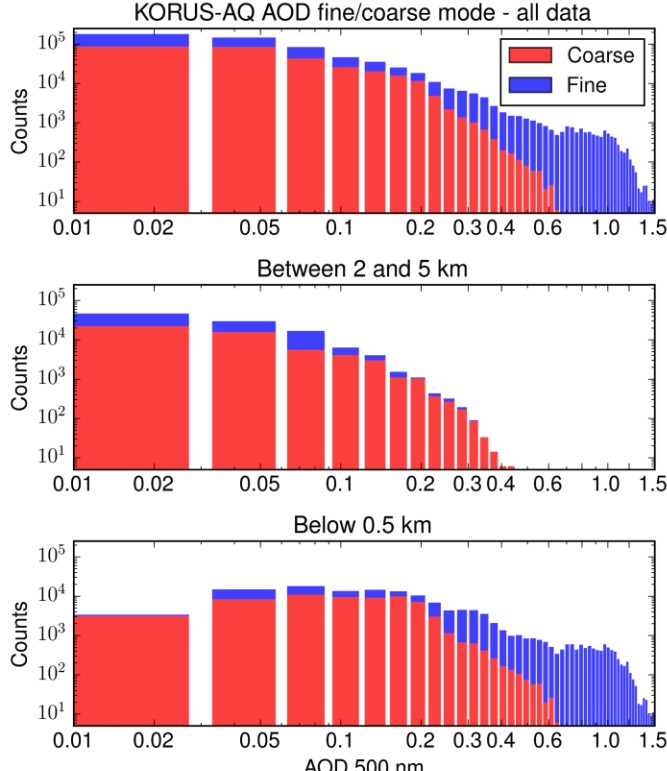

**Figure 8 - Histograms of fine and coarse mode AOD measured during KORUS-AQ for all data (top), AOD measured between 2 km and 5 km (middle), and below 500 m (bottom). The total length of the histogram bar indicates the total AOD in that bin, while the red/blue differentiation indicates the portion of the total AOD due to AOD from either fine or coarse mode aerosol.**


During the multiple-month deployment, the relationship between fine and coarse mode aerosols at different altitudes shifted with time and changing meteorological periods. The proportion of the impact of fine and coarse mode aerosols on AOD is also illustrated with AE (Fig. 9), which is inversely proportional to size. Notably, the dynamic meteorological regime showcases the largest AE at the highest altitude (smaller particles than other periods), while

the smallest AE is observed at high altitudes during the extreme pollution / transport meteorological period, which coincides with long range transport of aerosol. The low AE at high altitude suggests that the largest particles are transported, which supports the back-trajectories and meteorological estimate from Peterson et al. (2019) showing transport from northeast during this timeframe. The dynamic period shows a large variation in AE at higher altitudes, but may still be influenced by dust emissions, which have been shown to have relative variations on AE dependent on

dust layer-height, related to the transport pathways (Shin et al., 2015). The extreme pollution / transport regime shows a stratification of the aerosol layer for small AEs at lower altitudes (below 2 km) than all the other periods, however the FMF for that same time period and vertical region is highest than all other observations. This supports the observations by Eck et al. (2020) that a larger peak of fine mode is present during this period, relating to growth of small particles due to humidification or cloud-processing. Other notable features are during the blocking period, the

AE is similar to the overall averages from 2.5 km upward, while diverging to a higher AE in the lower portion of the atmosphere than the KORUS-AQ average. The largest interquartile spread in binned AEs is observed in the range between 1.5 km to 3 km, while the dynamic meteorological period represents the largest interquartile range for most

of the upper atmospheric observations (above 3 km). Similar vertical dependence as separated by meteorological periods has been shown by measurement of small aerosol particles acting as cloud condensation nuclei (Park et al., 530    2020a).

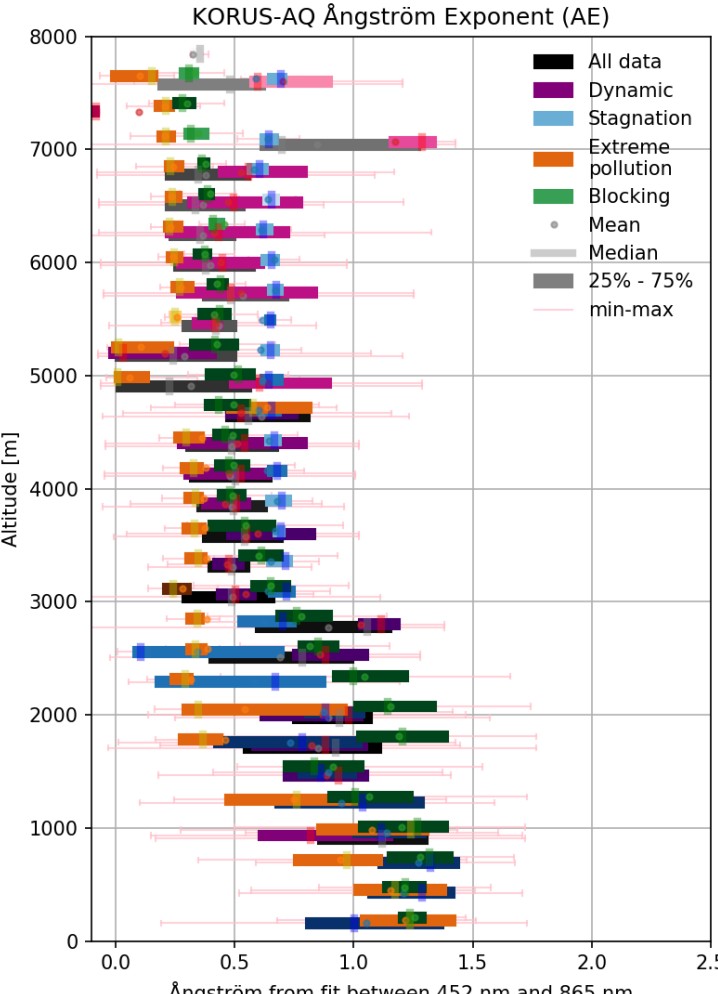

**Figure 9 - Ångström Exponent (AE) box-plot distribution measured at different altitudes separated by different meteorological periods, during KORUS-AQ, and for all data (black). The number of days sampled within each vertical bin and meteorological regime is illustrated by the shading of the color. The vertical bar indicates the median of each altitude bin, while the center dot represents the mean. The interquartile range of the data in each vertical bin is represented by the thick horizontal bar. The range of values observed at the given altitude is presented as horizontal, colored, error bars.**

### 4.4 Autocorrelation distances of intensive and extensive aerosol properties

Aerosols in the Korean peninsula region are subject to processes linked to their sources, sinks, and evolution. For aerosols subject to relatively minor evolution (e.g., dust with no photochemical aging and few removal processes), but 540    large transport distances, the autocorrelation distances are commensurate to the transport distances, and one would expect the inverse to be also true. When considering aerosol transport, the intensive properties are expected to remain constant, such as size, mass absorption efficiency, and index of refraction, but the total concentration within a column, impacting the AOD, would change due to dilution and removal of the aerosol (e.g. via rain out or dry deposition). For

example, dust aerosol transported from mainland China, which after initial growth with chemical and morphology changes, by coagulation and condensation, near source or after cloud/fog processing and humidification/dehumidification, have near constant intensive properties but experience dilution causing reduction in AOD, but no change in spectral dependence. This same dust transported from mainland China may experience external mixing with the fresher pollution from Korea (Heim et al., 2020), would both impact the AOD and its the spectral dependence at scales commensurate to the mixing region. Alternatively, for local aerosol production and growth, both

the intensive properties and the extensive column aggregate properties, like the AOD, will vary within small distances, akin to the size of the source region and the rate of the secondary organic aerosol production. Even advected aerosols from the surrounding region will undergo local processing, such as hygroscopic growth, particularly in Korea for ammonium sulfate, ammonium nitrate, and organic aerosols (Saide et al., 2020), which impact intensive properties (size and AE) within a small distance, while simultaneously increasing the AOD extensive property. Figure 9 shows

the autocorrelation as a function of distance of the aerosol properties measured and modeled during KORUS-AQ, along horizontal flight segments of the NASA DC-8 where there are 4STAR observations that are quality assured. The altitudes of these segments are presented in Fig. 7c.

The extensive aerosol property investigated here is column AOD, and the intensive aerosol property is AE, which is inversely proportional to aerosol size and dependent on aerosol refractive index (e.g., Saide et al., 2020). The

autocorrelation-distance distribution of AE is nearly identical to the FMF of the aerosol for the remotely sensed products. All data points used to build these relationships were first matched in time and location to the NASA DC-8 horizontal flight segments.

From Fig. 10, we see that the extensive aerosol property, AOD, has a high correlation, over longer distances (e.g., R>0.85 for distances up 25 km from 4STAR) than the intensive aerosol property, AE, (R>0.85 for distances up to 7.5

km from 4STAR). This difference between a consistently high autocorrelation over longer distances is particularly evident in the in situ data, but is reproducible with all observations/model. This is partly counterintuitive to the general notion (as described by Anderson et al., 2003) that aerosols have more consistent intensive properties from particular point sources, than their extensive counterparts. We find here that aerosol concentration is less variable than their size. While there are industrial point sources west of Seoul, most sources of aerosol impacting air quality are due from

diffuse (e.g., secondary formation from traffic and transport emissions) or from long range transport. From these samples during KORUS-AQ, the AOD is more consistent over a greater area than aerosol size, and consequently type. This may be linked to the notion of AOD and aerosol concentration is regulated mostly by the combination of long-range transport and changes in local sources, which are in turn modulated by the meteorological periods (e.g., Peterson et al., 2019). The local aerosol production and aerosol evolution/transformation may be more related to the underlying

processes and changes in dominant aerosol types, that impact aerosol at shorter distances, and consequently timescales, than the transport process (Heim et al., 2020).

The shortest autocorrelation distances are subject to both the random noise from instruments and retrievals, and the natural variability of the observed physical property (Anderson et al., 2003). The best-case scenario is to have

autocorrelation values near 1.0 for the shortest distances, indicating low natural variability and low noise from the

observations/models. In this comparison, since the observable quantity is consistent, the same natural variations should be present for all of the observations/models, therefore any reduction in autocorrelation can be attributed to the method's smallest observable distances and source of random noise. For sake of comparison of the physical processes, we opt to mostly ignore the autocorrelation at the shortest distances except to serve as a baseline upper-bound of the autocorrelation that can be resolved by MERRA-2, 4STAR, GOCI, and in situ sampling. In this comparison (Fig. 10),

both MERRA-2 and 4STAR have near 1.0 autocorrelation at the shortest distances, while GOCI and the in situ observations have lower values. This high initial value in autocorrelation can be interpreted as an upper bound of autocorrelation values that can be resolved by those methods. While MERRA-2 is interpolated to match 4STAR sampling, the native pixel resolution is still at roughly 59 km, thus results referring to distances shorter than that may be more indicative of the interpolation methods (Collow, et al., 2020) than the native modelling processes.

For all data observed and modeled during KORUS-AQ, the AOD at 500 nm wavelength ($AOD_{500}$) shows the longest distances with high autocorrelation, as compared to the AE (Fig. 10). The absolute magnitude of the autocorrelation for the observations and model is not as instructive as the relative decrease with distance. In both the AOD and AE autocorrelation, we include the measure from the in situ aerosol extinction coefficient, representing the aircraft-level measurement. As expected, because of the integrating effect of the column values and the increased sampling volume,

4STAR, MERRA-2, and GOCI values have longer distances with high autocorrelation, while the in situ aerosol extinction coefficient (AOD equivalent for point measurements) and AE have decreased autocorrelations at shorter distances. However, at longer autocorrelation distances, the autocorrelation of AOD (at >20 km) and AE (at >2 km) from GOCI mirror those from the point-like in situ measurements of the aerosol extinction coefficient.

To account for potential sampling biases, a subset of 91 segments (30%) of the total 302 horizontal flight segments,

comprising 583,183 samples, were randomly selected via Monte Carlo sampling, and repeated for building a 50-member ensemble. The ensemble mean and standard deviation are interpreted as the potential impact due to changing the selection of samples (Shinozuka and Redemann 2011). We observed relatively small autocorrelation divergence from the majority of dataset subsamples at the shortest distances (represented by the vertical error bars, Fig. 10). The standard deviation of the Monte Carlo sampling is largest for the longest distances, where fewer of the horizontal

segments span that length. The standard deviation of the ensemble sampling is smallest on average for AE than AOD, for all observations and models except for 4STAR derived values, with MERRA-2 showing the least dependence on sampling biases. The larger deviation of 4STAR AE than AOD within the Monte Carlo sampling is because AE is more variable and the AOD as measured by 4STAR, this may be caused by smaller AE range is available for GOCI or MERRA-2 due to their confined number of aerosol microphysical models, and that the in situ observations may be

limited to a subset of the aerosol due only sampling aerosol at the aircraft level. 4STAR's column measurement may also reflect the influence of multiple aerosol sources and types (e.g., dust over pollution) that are not represented in modeled, retrieved, or at-aircraft-level in situ measurements, some of that variation can be observed in Fig. 9.

Both MERRA-2 and 4STAR show similar autocorrelation over a wide distance range for AOD, with the 85[th] percentile point (where autocorrelation is reduced by 15%) occurring at 35 - 100 km (Fig. 10). Even though MERRA-2 uses assimilation to link its model representation to the observed world using MODIS and other remote sensors, it still shows the longest distance with a consistently high autocorrelation. Even when accounting for the 50-member standard deviation the MERRA-2 AOD autocorrelation only overlaps with the 4STAR AOD mean-member at distances longer than 60 km. For AE from both MERRA-2 and 4STAR, the standard deviation and mean of the member ensemble do not overlap until distances of >60 km, with MERRA-2 showing consistently higher autocorrelation than observed by 4STAR. The GOCI observations show a much shorter autocorrelation distance at the 85[th] percentile for AOD, just shy of 100 km, at which the overall trend follows the point-like in situ observations by LARGE. This is observed with both the AOD and AE autocorrelations. The AE autocorrelation decreases at shorter distances compared to AOD for all samplings (MERRA-2, 4STAR, GOCI, and in situ), with much higher downward slopes. MERRA-2 AE also shows a distinct inflexion in autocorrelation at 35 to 65 km, while 4STAR, GOCI, and in situ all show a constant steeper slope. The average distance to decrease autocorrelation by 15% for AOD of all methods is 52.5 km, for AE it is 22.1 km.

The 15% decrease metric is used to identify where there is an inflection point in autocorrelation, however the distances at which autocorrelation decays by 10% or by 1/e show similar trends (see Table 2 for examples). For AOD, the mean distance where there is 10% decrease in autocorrelation occurs at 26.8 km, roughly 1/2 the distance of 15% decrease, while an 1/e decrease (~37% reduction), the mean distance is 167.5 km. Because of the larger dependence to samples (showing larger spread in autocorrelation at longer distances), the 1/e results in a larger spread of distances (standard deviation of 54 km). Similarly for AE, where the largest standard deviation and spread is found from the 1/e decrease level.

These results contrast to those reported by ground-based observations from AERONET during KORUS-AQ, as presented by Choi et al., (2021), which shows smaller changes in FMF than AOD for coarse or fine mode as a function of distance between the AERONET ground sites (0.11/100 km for FMF, 0.16/100 km, and 0.14/100 km for AOD fine and coarse mode). However, Choi et al. (2021) also shows a lower correlation in FMF, and arguably, a non-linear relationship, particularly at distances shorter than 100 km. This non-linear relationship is presented here in Fig. 10.

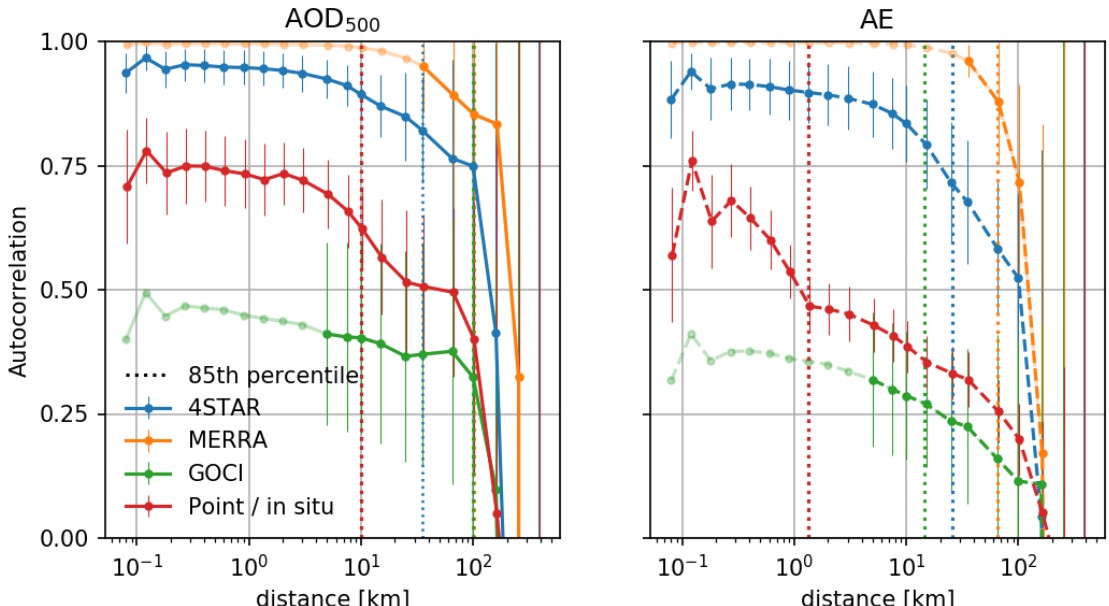

**Figure 10 - The autocorrelation distances of the aerosol properties measured and modeled during KORUS-AQ, separated by either aerosol intensive properties (right, AE, relating to aerosol size, dashed lines) or extensive properties (left, column AOD, solid lines). The point / in situ denotes the autocorrelation from the aerosol extinction coefficient instead of the column values of all others. The colors denote the source of the data presented here. The vertical error bars represent the standard deviation of the 50-member ensemble Monte Carlo subsampling at 30% of flight segments. The vertical dotted lines represent the autocorrelation distance of the 85th percentile, which is the location that the autocorrelation is reduced by 15% when compared to the autocorrelation of the shortest distance. The lighter colored lines for GOCI and MERRA-2 represent the autocorrelation for distances smaller than the diagonal length from one pixel center to another, which are closest to the NASA DC-8 flight path.**

The distances at which autocorrelation varies can also be understood through the expected variation of the aerosol properties (AOD or AE). Since we use the combination of all level flight legs, the difference between AOD or AE between measurements binned by their lag distance has a mean and median very close to zero, while the standard deviation grows with distance. The near-zero mean difference in AOD or AE at varying distances implies an even distribution of measurements. Table 2 shows the values of the mean, median, and standard deviation for AOD and AE at distances with different autocorrelation reduction. Notably at 90% of relative autocorrelation, the standard deviation in AE from 4STAR exceeds by 0.048 the median uncertainty (see Fig. 4b) expected for the average AOD value of 0.36, or even for 63% of the measurements below 1000 m.

| Relative auto-correlation | AOD$_{500}$ | | | | | AE | | | | |
|---|---|---|---|---|---|---|---|---|---|---|
| | 4STAR | MERRA-2 | GOCI | in situ (LARGE) | Average | 4STAR | MERRA-2 | GOCI | in situ (LARGE) | Average |
| **90%** | | | | | | | | | | |
| distance [km] | 25 [10, 35] | 65 [35,65] | 10 [3,160] | 7.5 [7.5, 7.5] | **26.88** | 10 [10, 15] | 65 [35, 100] | 0.6 [0.27,1.35] | 0.27 [0.2, 0.27] | **18.34** |
| mean of difference | 0.003 | -0.0015 | 0.008 | 0.013* | **0.003** | -0.0036 | -0.0161 | -0.00098 | 0.0015 | **-0.005** |
| standard deviation | 0.1364 | 0.0504 | 0.0985 | 0.292* | **0.095** | 0.1465 | 0.206 | 0.3072 | 0.1776 | **0.209** |
| **85%** | | | | | | | | | | |
| distance [km] | 35 [25, 35] | 100 [35,100] | 65 [5, 160] | 10 [10,10] | **52.5** | 15 [15,25] | 65 [35,160] | 7.5 [5,7.5] | 0.9 [0.6,0.9] | **22.1** |
| mean of difference | -0.0072 | -0.0073 | 0.0185 | 0.0173* | **0.001** | -0.0039 | -0.0161 | 0.0027 | 0.0048 | **-0.003** |
| standard deviation | 0.1436 | 0.0682 | 0.1442 | 0.318* | **0.119** | 0.162 | 0.2066 | 0.408 | 0.217 | **0.248** |
| **1/e (63%)** | | | | | | | | | | |
| distance [km] | 160 [35,160] | 250 [35,250] | 160 [65, 160] | 100 [35, 100] | **167.5** | 100 [100,100] | 160 [100, 160] | 100 [25, 100] | 65 [65, 100] | **106.25** |
| mean of difference | -0.0073 | -0.0985 | -0.033 | 0.0057* | **-0.046** | 0.037 | -0.065 | 0.028 | 0.037 | **0.009** |
| standard deviation | 0.1557 | 0.1357 | 0.217 | 0.389* | **0.169** | 0.247 | 0.354 | 0.507 | 0.371 | **0.370** |

**Table 2 - Distance bins at which different relative autocorrelation is reached for AOD and AE from 4STAR, MERRA-2, GOCI, and in situ (LARGE). The range in distance (square brackets) is obtained from the autocorrelations that are varied by one standard deviation of the 50-member Monte Carlo ensemble of flight segments. The differences in AOD and AE from all flight segments at the distance bins are reported by their mean and standard deviation. The Average AOD mean and standard deviation of the difference is averaged from 4STAR, MERRA-2, and GOCI, while AE also includes in situ (LARGE). *The in situ extinction coefficient mean difference and standard deviation are multiplied by 2.5 km for easier comparison to the AOD mean and standard deviation values.**

What remains to be clarified are the differences between our current understanding of aerosol point-sources and their combined impact on meteorology, vertical distribution of aerosols, and aerosol speciation.

**4.5 Untangling the impact of meteorology, altitude and speciation on autocorrelation**

The average AOD during KORUS-AQ is dependent on the meteorological regime (Fig. 3) and the vertical sampling (Fig. 7). The AOD vertical distribution of fine and coarse mode aerosols (Fig. 8) is a driver for changes of AE as a function of altitude and combined together for the total column. However, the AE vertical distribution is also dependent on the different meteorological periods (Fig. 9). Here we show the autocorrelation distances of the AOD and FMF, as a function of meteorological periods and sampling altitude for a variety of observations and models (Fig. 11). For easier comparison we opted to show the autocorrelation normalized to the shortest distance, since reduction in autocorrelation over varying distances was the main focus, and less the resolution of the instrumental noise. Additionally, the autocorrelation distances as reported by Shinozuka and Redemann (2011) for the long-range

transport from Arctic observations and local biomass burning in the Canadian Boreal Forest are included for reference (SR2011 Long, SR2011 Local; respectively).

Random sampling of the level flight segments allows illustration of the range by which the autocorrelation depends on the specific flight segments. The most variability is observed at the longest distances where there are fewer samples

from the flight segments (Fig.11). Significant changes between the autocorrelations are observed for the varying meteorological periods, with the blocking period having the shortest distance which was negatively correlated with both the AOD and FMF. The blocking period, which experienced a high-pressure ridge diverting much of the mid-latitude storm tracks away from Korea but with the highest average surface temperature and with most days having a cloud fraction over 50%, is also the only period when the distance to 85$^{th}$ percentile autocorrelation for MERRA-2

AOD is the same as for 4STAR AOD. This last time period, may be subject to largest variations in both AOD and FMF due to rapid aerosol growth by water vapor condensation on the aerosol particles, and amplification of secondary aerosol formation, that occur in the warm and humid Korean summertime (Koo et al., 2021; Jordan et al., 2020).

The meteorological period with the preceding longest distance at high autocorrelation was the dynamic period. While this is true for both AOD and FMF, the FMF behaves less monotonically than AOD autocorrelation, with larger

variations with the Monte Carlo subsampling that encompasses different flight legs with potentially different aerosol composition. The largest difference between autocorrelation of AOD from 4STAR and MERRA-2 occurs during the stagnation period, which had a significant anticyclone flow over Korea. This would suggest that MERRA-2 either undervalues the removal processes of the aerosol during that period, or that the AOD sources are less consistent than what is expected by the model. The stagnation period, which was a particularly dry but hot period, also has the second

shortest distance with high autocorrelation of FMF (repeated by both 4STAR and GOCI), indicating a high variability of the aerosol size during this time, albeit with higher uncertainty in FMF and AE measurements due to low AOD. The dry conditions were unlikely to be favorable for secondary organic aerosol formation (Liu et al., 2018), but the heat of the day contributed to mixing depths of 1500-3000 m (Peterson et al., 2019), which would point to changes in aerosol removal processes in MERRA-2 rather than sources. In tandem, the ebb and flow of sea breeze shifted aerosol

from over the yellow sea and Seoul, likely contributed to variations in aerosol during this time period (Eck et al., 2020). However persistent clearer skies, obscured by some dust but less clouds, may promote photochemistry in tandem with the inflow of ozone from sea breeze near Seoul. This period is also identified has having significant changes in vertical mixing, which in turn impacts the containment and distribution of aerosol (Jordan et al., 2020). The representation of the vertical mixing depth in MERRA-2 could be influential to its autocorrelation representation.

For all periods, GOCI presented lower autocorrelation than 4STAR for both AOD and FMF, and notably going lower during the blocking period than the reference SR2011 Local, which focused on wildfire/biomass burning events in Canada. This may be explained by the selection criteria for GOCI AOD retrieval, which returns the mean of the 3 best aerosol model (Choi et al., 2018), and at the same time accounts for variations in surface albedo.

While mostly following the same autocorrelation trends with distance as 4STAR AOD and FMF, GOCI AOD and

FMF have higher autocorrelation during the extreme pollution/transport period than the dynamic period which boasts

the highest autocorrelation for 4STAR and MERRA-2. These consistently high autocorrelation over large distances for the extreme pollution and dynamic periods (12.6 km and 15.2 km, respectively, with r(AOD) above 0.85 from 4STAR) in combination with the low average AE, matches the expectation of dust aerosol transported from long distances (Peterson et al. 2019). Additionally, these aerosols were identified as dust mix type because of their high depolarization ratio in the 3 km to 7 km range using the DIAL (Differential Absorption Lidar) / HSRL (High Spectral Resolution Lidar) measurements for that time period (KORUS-AQ Science Team 2019; HSRL - DIAL KORUS-AQ Flight 19 - May 30, 2016, accessed 2022). For these periods, the autocorrelation remains higher for longer distances for the FMF than the AOD, likely indicating that transformation of the aerosol size is not significantly affected during transboundary transport, e.g. no large spatial variations in rain out, new particle formation, or particle growth. There are increased ground-based observations of the small PM2.5 particles potentially linked to new-particle formation (e.g., Eck et al., 2020), however these are either not observed here, or not varying the FMF as much as the AOD. While there are evidence of pollution aggregating on dust particles (Heim et al., 2020) and large fine mode particles occurring during high RH and cloud fractions suggesting cloud processing or particle growth by humidification (Eck et al., 2020), this would be less impactful to the autocorrelation than aerosol processes occurring during the blocking period. The blocking period is notable because of the shortest distance with high autocorrelation for the FMF (3.5 km with r(FMF) greater than 0.85 observed by 4STAR), likely resulting from a variable combination of secondary aerosol formation, deeper vertical mixing from episodic and cloud processing of the aerosol similarly to PM2.5 (Jordan et al., 2020). These processes would result in rapid change in particle size, and thus shorter distances with high autocorrelation. Since these processes occur regionally for distributed sources on the Korean peninsula, we find that the distances of aerosol size change is smaller than those for change in optical depth (8.0 km with r(AOD) greater than 0.85 observed by 4STAR). This is likely from a combination of factors, including AOD from new particle formation more than compensates for aerosol dilution, similarly to that found downwind of Canadian oil sand processing centers (Baibakov et al., 2021). For the distances reported, there are no periods exhibiting higher autocorrelation than those sampled from long-range transport in the arctic (SR2011 Long).

The autocorrelations for both AOD and FMF, as segregated by altitude, have nearly indistinguishable behavior for distances shorter than 10 km. The altitude dependence of the autocorrelations shows that the highest sampling (greater than 3 km altitude, from all 20 flights) had the longest autocorrelation, as expected, likely due to the being affected mostly by long-range transport than local sources (Fig. 11b). The aerosol observed at the mid-layer (1-3 km, sampled by 18 of the 20 flights) has the shortest distances with high autocorrelation, while the lowest layer (below 1 km, sampled by 18 of the 20 flights) has its autocorrelation between the mid-layer and high layer, nearly identical to the average of all segments. The vertical mixing height was predominantly around 1 km during the dynamic and transport periods, but increased to nearly 3 km at Seoul during periods coinciding with shorter distances of high autocorrelation (stagnant and blocking) (Peterson et al., 2019; Jordan et al., 2020). The reduction in autocorrelation is likely a result of the changes in mixing layer height. The FMF for all altitudes had a more gradual decrease in autocorrelation over the whole range of distances than for AOD, while the autocorrelation for AOD dropped more precipitously. Consistent with Fig.10, the GOCI AOD and FMF have lower autocorrelations for set distances than the 4STAR and MERRA-2 counterparts.

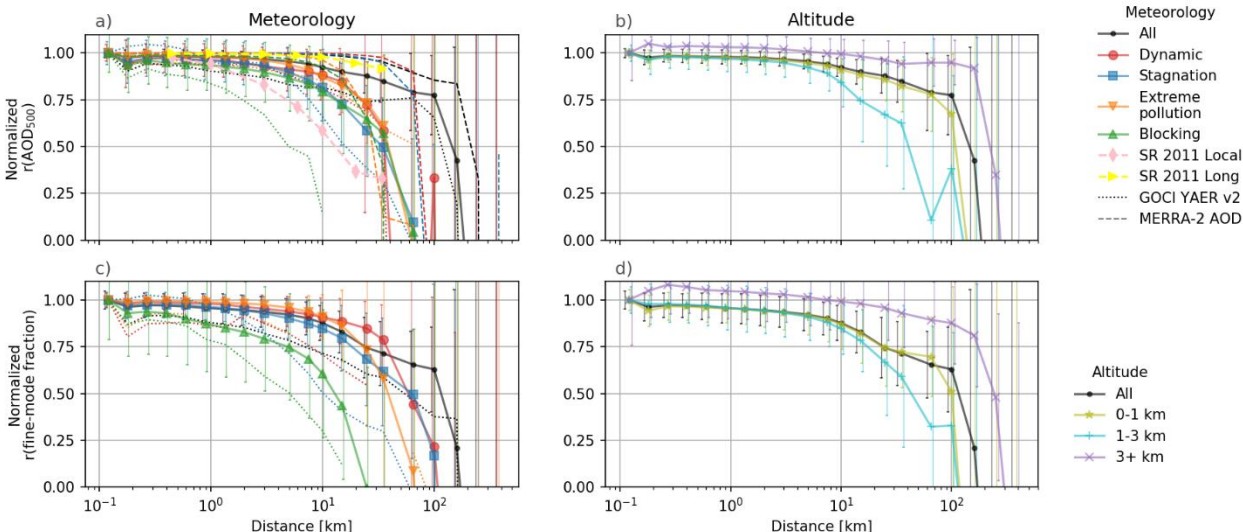

**Figure 11 - Autocorrelation of aerosol properties during KORUS-AQ, separated by meteorological time periods on the left-hand panels, and separated by altitude on the right-hand panels, with top panels representing the autocorrelation of extensive AOD properties and the bottom panels representing the intensive properties. The vertical error bars indicate the standard deviation of the autocorrelation computed using a 50-member Monte Carlo ensemble.**

Figure 12 shows the distances that the autocorrelation for speciated AOD were reduced by 15%, or at the 85[th]

percentile of the first autocorrelation value (as represented by the vertical dotted lines in Fig. 11). These 85[th] percentile

distances are the median values of the 50-member ensemble. We represent the 4STAR and GOCI AODs as separated

by their portions due to either fine mode or coarse mode aerosols. Including the MERRA-2 AOD enables

understanding of the AOD contributions from multiple aerosol types: dust and sea salt compared to optically defined

coarse mode aerosol (top panel Fig. 12), as well as sulfates, black, and organic carbon aerosols compared to the fine

mode aerosol (bottom panel Fig. 12). The autocorrelation distance is calculated based on data from the different

meteorological periods (all at altitudes below 3000 m).

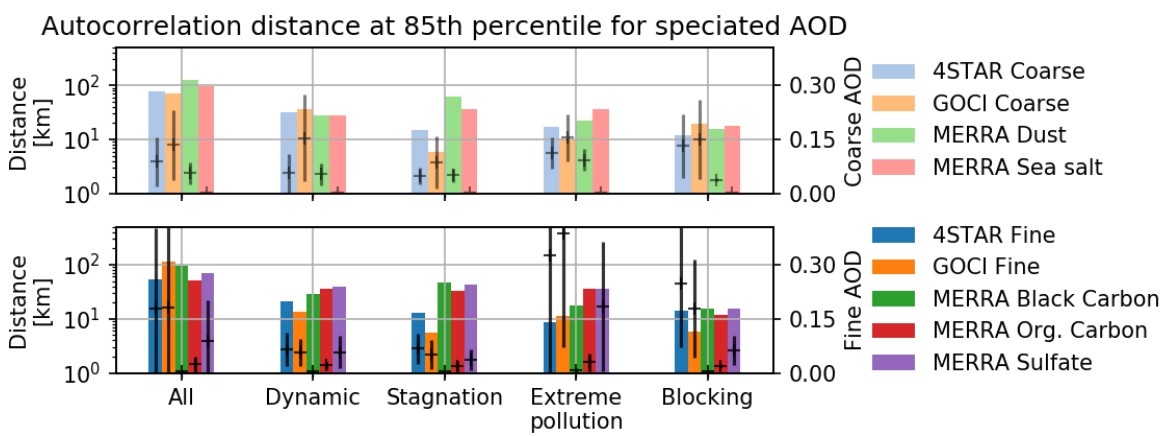

**Figure 12 - The distance for the 85[th] percentile autocorrelation, or which the autocorrelation is decreased by 15%, for a variety of speciated AODs, including fine and coarse mode AOD for 4STAR and GOCI, and dust, sea salt, black carbon**

**organic carbon, and sulfate contributions to AOD for MERRA-2. These are separated by the meteorological periods. The black symbols for each aerosol species indicate the average AOD, and the standard deviation (as error bars), with respect to the right-hand side y-axis.**

Of all the meteorological periods observed during KORUS-AQ, the dynamic period (first sixteen days of May 2016), showcased the largest distances with autocorrelations higher than 85%, and consistently amongst model, satellite, and

4STAR observations, even when discretized between various aerosol types, at 31.5 km ±2.7 km (20% bin size). The stagnation period, while having much lower average AOD, still showcased very similar 85[th] percentile autocorrelation distances, except for GOCI, which showed longer distances than during the blocking period, which had more AOD from sulfate aerosol as reported from MERRA-2. The distance of autocorrelation at 85% for fine mode AOD for the combination of all KORUS-AQ demonstrates a high degree of consistency between 4STAR, GOCI, and MERRA-2

sulfate, and, to a lesser degree, organic and black carbon AODs (Fig. 12). The extreme pollution meteorological pattern boasts the lowest 85[th] percentile with autocorrelation distances from all meteorological patterns, and the coarse mode having shorter distances than the fine mode. During this time period, the greatest proportion of AOD is attributed to sulfates by MERRA-2, with similar autocorrelation distances as determined by 4STAR, while the GOCI average AOD is overestimated during that time period. The 85[th] percentile distance for the fine mode AOD from 4STAR seems to

be consistently shorter than modeled by MERRA-2, except for the AOD due to black carbon, which also boasts a lower average AOD (black horizontal line, Fig. 12). When comparing the mean AOD from MERRA-2 by species to the different periods, the same trends (for aerosol mass density) appear of the chemical composition of particle matter of less than 1 micron aerodynamic diameters as measured at a ground site in Seoul (Jordan et al., 2020). While not exactly a one-to-one comparison, the sulfate AOD during the blocking period is greater than the average, while the

reported mass density for sulfates is slightly lower than the average at the surface during that same period.

## 5 Conclusion

The AOD measured during KORUS-AQ by airborne sampling using 4STAR, satellite remote sensing using GOCI, and reanalysis from MERRA-2 was found to follow general climatological trends for the Korean peninsula (Choi and Ghim, 2021). The aerosol intensive properties are also observed during KORUS-AQ, particularly aerosol size (fine or

coarse mode) and related AE. We present the general trends in AOD and AE/FMF sampled over the duration of KORUS-AQ, and show the vertical dependence, the impact of meteorological periods, and the autocorrelation as a function of distance for level flight legs. The spatial distribution of AOD was mostly matched between 4STAR, GOCI and MERRA-2, with the highest AOD in the Yellow Sea and near Seoul, and the lowest AOD observed just south and also directly east of the Korean peninsula. The uncertainty in AE and FMF were also evaluated and shown to peak at

low AOD, while the AOD uncertainty was more variable at larger AODs. Comparing the sampling by the NASA DC-8 to GOCI and MERRA-2 regional averages and subsets of observations matched to the NASA DC-8 flight paths, we observed that (i) 4STAR AOD was representative of the regional average and variability for 18 out of 20 days, (ii) 4STAR FMF has a high bias compared to the subsets of GOCI, and (iii) 4STAR AE has a low bias compared to MERRA-2. The relatively low GOCI FMF compared to 4STAR corroborates with the findings from Choi et al. (2018)

when comparing to AERONET sites.

The highest AODs were observed during the extreme pollution period (25-31 May), where transport was observed alongside haze formation (Peterson et al., 2019). This high AOD period was also observed using ground-based AERONET sensors (Choi et al., 2021; Lee et al., 2018). This meteorological period had the lowest AE at high elevations, suggesting lofted dust or other coarse mode aerosol, consistent with back-trajectories and DIAL/HSRL
(Peterson et al., 2019; HSRL - DIAL KORUS-AQ Flight 19 - May 30, 2016, accessed 2022). When observing the distance at which the autocorrelation of the extensive (AOD) and intensive (FMF/AE) properties are reduced by 15%, the meteorological periods seem to be the primary drivers, with the extreme pollution period showing the smallest distance. While there are variations between 4STAR, GOCI, MERRA-2, and in situ measurements, this shortest distance is observed during the extreme pollution, with the intensive properties showing shorter distances for a 15%
decrease of autocorrelation of extensive properties. With this autocorrelation decrease, there is an increase in standard deviation, which is larger than the median uncertainty of AE for the majority of samples.

During the stagnation meteorological period (17-22 May), we observed the lowest AOD (similarly to Choi et al., 2021), and the highest AE at elevated observations. The high AE is likely due to lofted small size aerosol. While this period showcased smaller aerosols in the column, it also hosted the longest distances at which the autocorrelation
remained above 85% at its lowest value. This long distance, was rivaled only by the blocking period (1-7 June), and was reproduced in all of our observation methods (4STAR, GOCI, MERRA-2, and in situ).

Throughout the entire period, the AOD due to fine mode aerosol constituted the largest contributor to AOD at lower altitudes (below 0.5 km), while the AODs due to coarse mode aerosol were predominant in the lofted vertical region (2 km - 5 km). The exact distribution of fine and coarse mode aerosols was modulated during the different
meteorological periods, with the altitude region between 1.5 km and 3 km showing the largest variability in FMF. Throughout the entire period, the distance at which a decrease of autocorrelation is observed to be consistently shorter for intensive properties than extensive, repeatable with 4STAR, GOCI, and MERRA-2. To account for potential sampling bias, we computed this distance and the autocorrelation by using a Monte Carlo ensemble of the level flight legs, and reporting its mean and standard deviation. This suggests that, contrary to commonplace opinion, when
dealing with a region where there are aerosols from multiple sources, the intensive properties are not as consistent over long distances as the extensive aerosol properties. This work showcases that in some regions the spatial scale at which aerosol size varies is smaller than that for aerosol optical depth.

## Appendix A: Measurement quality and corrections

### A.1 4STAR Aerosol Optical Depth derivation

AOD is calculated using the inversion of direct solar transmittance measured by 4STAR while actively tracking the sun on board the NASA DC-8. The simple inversion process is based on the refined Beer's Law but is subject to multiple correction procedures, namely the correction of transmittance, based on the variability of transmittance influenced by the Fiber Optic Rotating Joint (FORJ), the non-linearity of the spectrometer, the removal of the trace gas column impact on the AOD spectra, and the correction of the transmittance change due to unwanted material
deposition on the 4STAR window, similarly described by LeBlanc et al. (2020).

### A.1.1 FORJ correction and gas phase optical depth

Sunlight entering the 4STAR Gershun tubes is propagated to the spectrometers using low-loss multimode optical fiber bundles and other fiber optic components. The FORJ is part of the light path that allows endless rotation of the 4STAR sun-tracking head azimuthal position, with respect to the fixed geometry of the aircraft fuselage, where the rest of the
4STAR instrument is located. The FORJ introduces variability, which includes angle dependent hysteresis and some random noise, in the transmission due to this azithumal position, and can be corrected. The correction is computed from the azimuthal dependence through measurements of a stable light source (a light emitting diode that has less than 0.1% variation in radiance during the time of the test) in between each flight by a full rotation in each direction. The variations have a near sinusoidal shape with features departing from the mean by no more than +/-1.4% and are
repeatable in between each measurement (within 0.4% over the course of the field mission), with the largest features not moving by more than 20 °.

AOD is influenced by trace gas absorption in the entire column in distinct wavelength regions. We correct the influence of trace gas ($NO_2$, $CO_2$, $O_3$, $O_2$-$O_2$, $CH_4$) by convolving their retrieved vertical column gas abundance and profile with their spectral absorption coefficients (Segal Rozenhaimer et al., 2014). This result in an optical depth
contribution from these gases (typically very minor) which is then subtracted from the AOD spectrum.

### A.1.2 Updated 4STAR light path instrument design for thermal stability

Prior to the KORUS-AQ deployment, the 4STAR flight path was improved using funding from the ESTO Airborne Instrument Technology Transition (AITT) program to reduce variability in the transmission of the optical path. Notable modifications include improved fabrication processes for the fiber optic assemblies, and improved polishing
and cleaning of the fiber optic ends and direct beam diffusing element to ensuring a flat field of view (less than 1% deviation over 1°) for sampling the solar direct beam.

Fiber optic assembly modifications include annealing of the PEEK jacketing to reduce the degree of dimensional creep induced by the considerable thermal cycling exposure, the use of thermal epoxy heat shrink tubing to reinforce the jacket-to-connector interfaces, and careful control of the fabrication geometry to ensure that there is sufficient radial
clearance in the coils to accommodate the thermal expansion differential between the fused silica fiber and the PEEK jacket, to avoid thermal-induced stresses and resultant microbending losses and connector reliability in the light path, for temperature ranges from -50°C to +50°C. The fiber optic ends have been repolished to ensure minimal light scattering and a high degree of flatness for each of the fiber optic bundle endfaces that link the spectrometers and the optical inlets (see Fig. A1, for before and after fiber optic bundle example). Additionally, procedures were adopted
for the preparation, curing, polishing, inspection and cleaning of the fiber optic connectors.

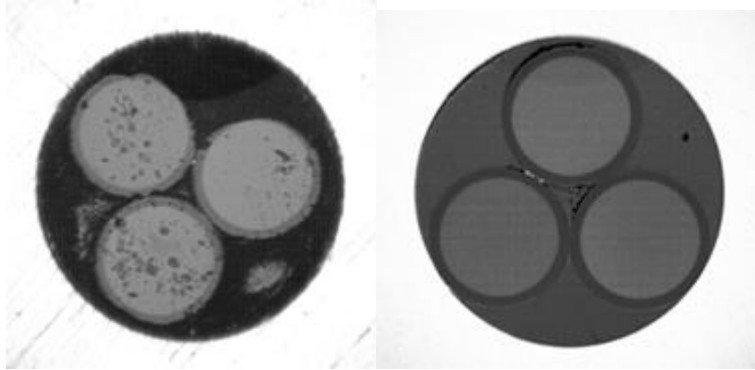

**Figure A1 - Image of 3-element fiber optic bundle endface that is part of the solar direct beam measurement optical path of 4STAR. Before polishing is seen on the left, while after polishing on the right. The uneven fiber flatness, severe contamination and pocked endface increased scattered light and was linked to distortions in the instrument field of view. The polished fiber face has a very planar endface, with no pocking and contamination resulting in a much more predictable light measurement.**


Prior to KORUS-AQ, 4STAR's direct irradiance measurement was dependent on the instrument head temperature, which had to be corrected. This impact was up to 0.15%/°C in transmittance. This was caused by the thermal expansion

of the metal ferrule at the end of the fiber optic assembly pressing against the Spectralon diffusing element at the base of the light collecting Gershun tube. Spacing between the fiber optic bundle face and the Spectralon diffuser was adjusted, and mechanically stabilized to prevent temperature-dependent pistoning of the connector from encroaching onto the diffuser.

Through cycling the head temperature in lab settings, the temperature dependence of the improved light path has been

identified to vary less than 0.004%/°C, which is roughly equivalent to the worst-case scenario of optical depth changes of less than 0.004 with the sun directly at zenith, for temperatures ranging from +50°C to -50°C. Typically the error in optical depth due to the temperature variation was about 0.002, due to the sun angle being on average much lower than zenith, and the majority of the temperature ranged to about half of the extremes.

## A.1.3 AOD window deposition correction

During KORUS-AQ, 4STAR was subjected to varying thermal and atmospheric conditions and polluted atmospheres. During the science flights of the DC-8, the 4STAR window exposed to ambient air at the top of the DC-8 became coated by a persistent thin contamination film that resulted in a reduction in the transmission efficiency of the window for potentially the remainder of the flight, until the window surface was cleaned when 4STAR was back on the ground.

Table A1 showcases the impact of window deposition as measured on the ground post-flights by measuring the change in signal from a stable light source before versus after cleaning the window, with missing elements representing negligible (less than 1%) impact. For 14 out the total 25 flights (research + check + transit flights), the amplitude difference was less than 2%.


| Date | Flight number | Difference [%] |
|---|---|---|
| 20160418 | PCF1 | - |
| 20160421 | PCF2 | - |
| 20160426 | TR1 | - |
| 20160427 | TR2 | 1.28 |
| 20160501 | 1 | 1.11 |
| 20160503 | 2 | - |
| 20160504 | 3 | - |
| 20160506 | 4 | 2.80 |
| 20160510 | 5 | - |
| 20160511 | 6 | 6.22 |
| 20160512 | 7 | 1.21 |
| 20160516 | 8 | 6.88 |
| 20160517 | 9 | 1.62 |
| 20160519 | 10 | - |
| 20160521 | 11 | 3.41 |
| 20160524 | 12 | 36.00 |
| 20160526 | 13 | 1.79 |
| 20160529 | 14 | 10.11 |
| 20160530 | 15 | 18.56 |
| 20160601 | 16 | 19.33 |
| 20160602 | 17 | 8.35 |
| 20160604 | 18 | 16.09 |
| 20160608 | 19 | 18.72 |
| 20160609 | 20 | 22.09 |
| 20160614 | tr3 | - |

**Table A1 - Magnitude of light intensity change due to cleaning 4STAR's window post-flight, for the highest magnitude peak of the LED light source near 650 nm wavelength. Research flights are numbered, while project check flights and transit flights are indicated by acronyms PCF and TR, respectively.**

To correct for window deposition on the AOD, each flight with greater than 2% post-flight light intensity difference is manually inspected to determine the set of discrete events leading to the deposition, notably during low-level near-water flight segments, in highly polluted periods, or during cloud insertions. The uncertainty in the AOD surrounding these events (within ±6 min) has been increased to the magnitude of the optical depth of the window deposition and by 30% of the corrected magnitude for the rest of the flight, producing a step-change in the AOD uncertainty. The full solar spectra measured at high altitude flight segments (above 6 km) are used to evaluate the change in spectral optical depth due to window deposition, where the AOD is expected to be mostly representative of the stratospheric aerosol, at around 0.03. Fortunately, the typical flight maneuvers required profiling through the boundary layer and up to higher altitudes, frequently, enabling a 'bookend' check on the window deposition, confirming a reasonable correction. Figure A2 presents the impact of the window deposition correction on the AOD at 501 nm for all the flights during KORUS-AQ, as a function of altitude. A near constant vs. altitude impact is shown here of up to 0.1, while the AOD vertical dependence is not highly impacted.

KORUS-AQ Aerosol properties spatial distribution

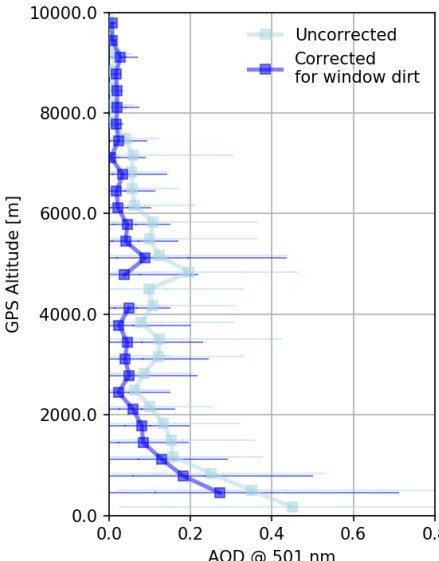

**Figure A2 - Summary of the impact on correcting the AOD at 501 nm for window deposition during KORUS-AQ, as a function of altitude. The AOD is binned in roughly 200 m altitudes, with the error bars representing the interquartile range for each of those bins.**

**Appendix B: AOD comparisons**

**B.1 4STAR-GOCI AOD comparison**

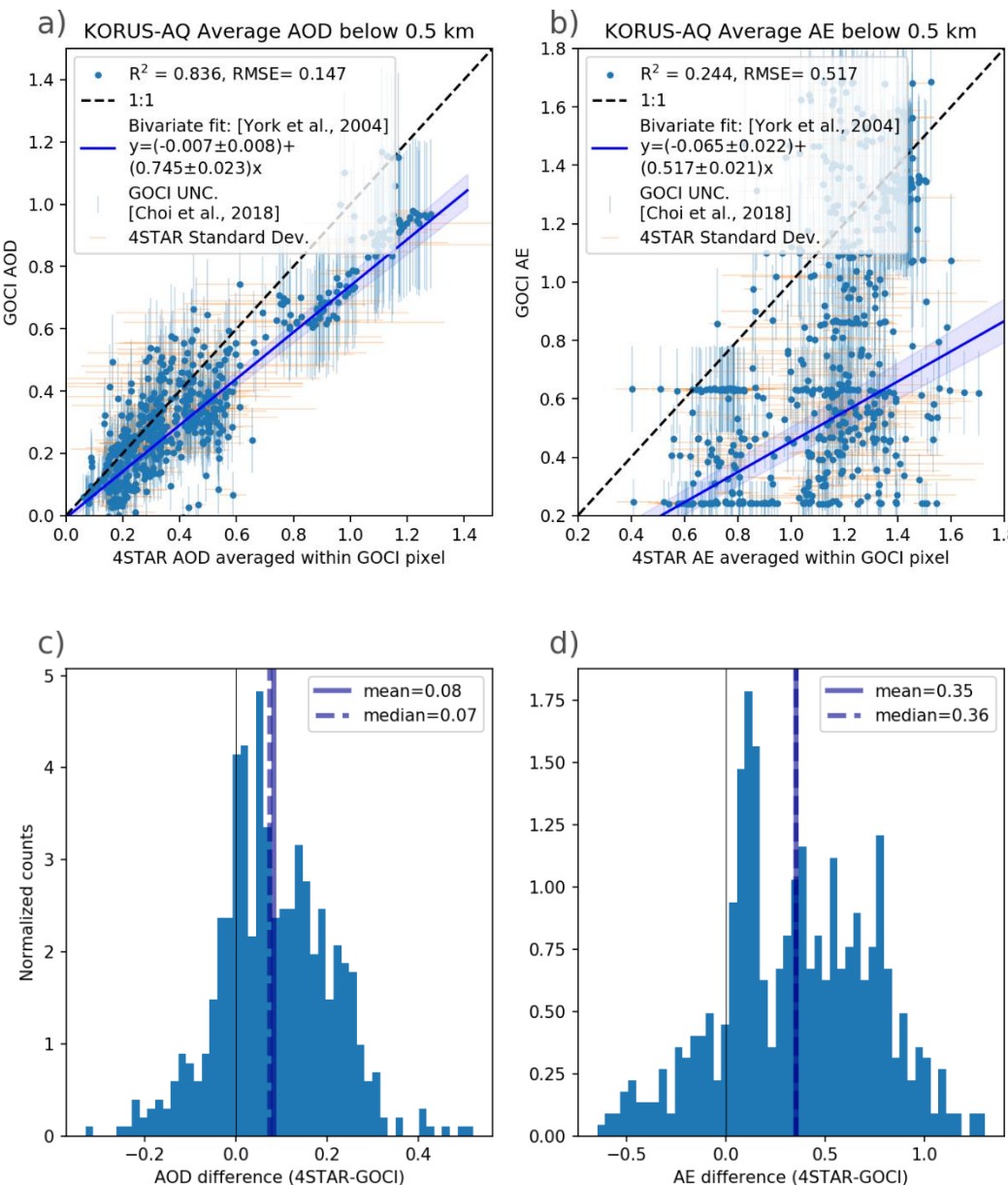

**Figure B1 - Comparison of AOD at 500 nm from GOCI retrievals as compared to 4STAR measurements below 500 m. For each cloud cleared GOCI pixel, the nearby 4STAR samples (within 30 minutes of GOCI retrievals) are averaged together for this comparison, shown in the upper pair of graphs (a for AOD and b for AE). The linear fit used here is the bivariate fit described by York et al. (2004), and reinforced for use by Cantrell (2008). The difference between the matched 4STAR to GOCI values are presented in the bottom pair of histograms (c for AOD and d for AE).**


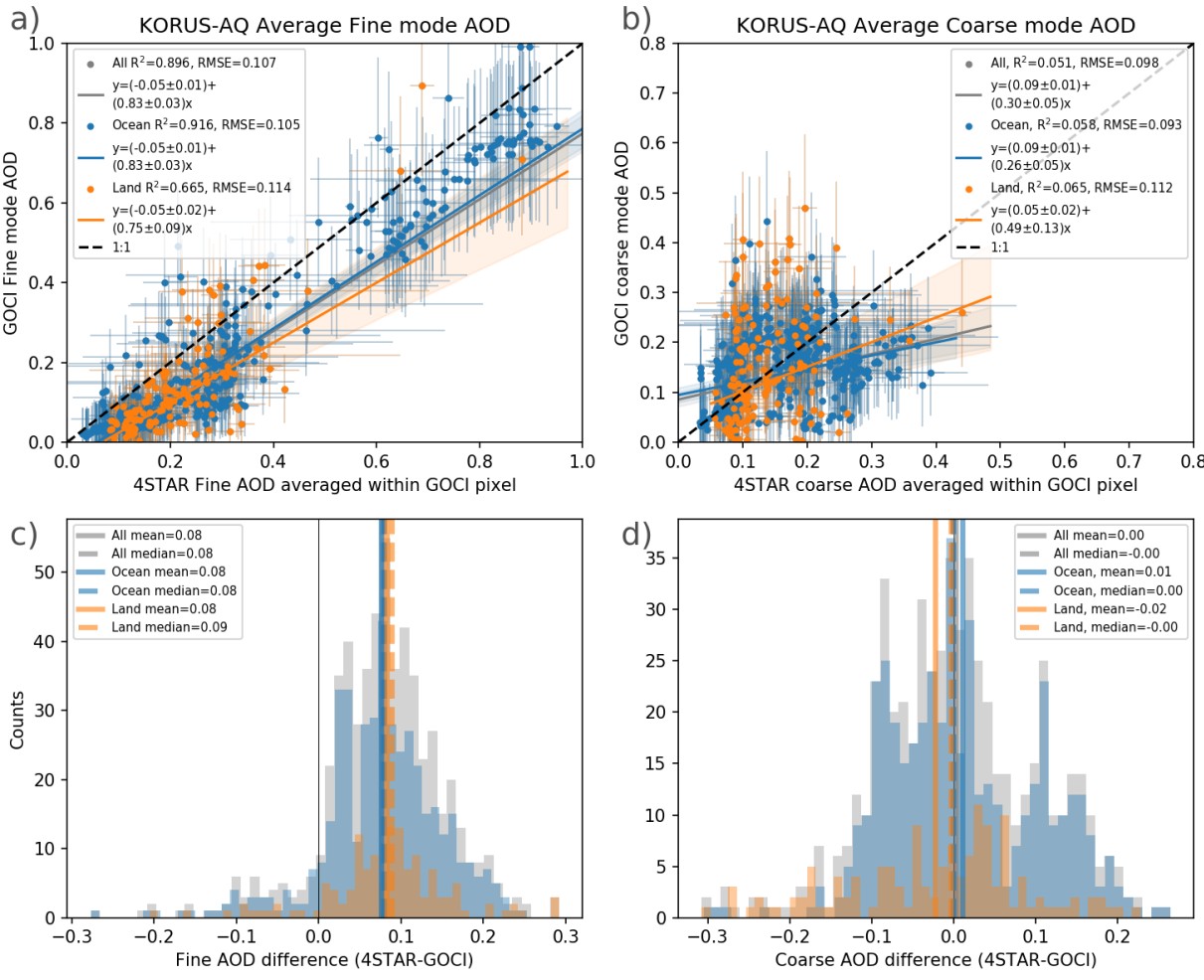

**Figure B2 - Average 4STAR (a) fine and (b) coarse mode AOD within the different GOCI pixels matched on the same observation times and locations for all measurements under 500 m. The uncertainty in GOCI is using those characterized by Choi et al. (2018). 4STAR fine and coarse mode AOD is the standard deviation within the matched GOCI pixels. The linear relationship is quantified using a bivariate fit described by York et al. (2004). The differences between the observations by 4STAR and GOCI is found for fine mode AOD in (c) and for coarse mode AOD in (d). This comparison is subset between AOD measured over land (in orange) and over the ocean (in blue). The separation between land and ocean was achieved using the 1-km resolution GLOBE data (Hastings and Dunbar, 1999) ported for use in python (Karin, 2020).**


**B.2 4STAR-MERRA-2 AOD comparison**

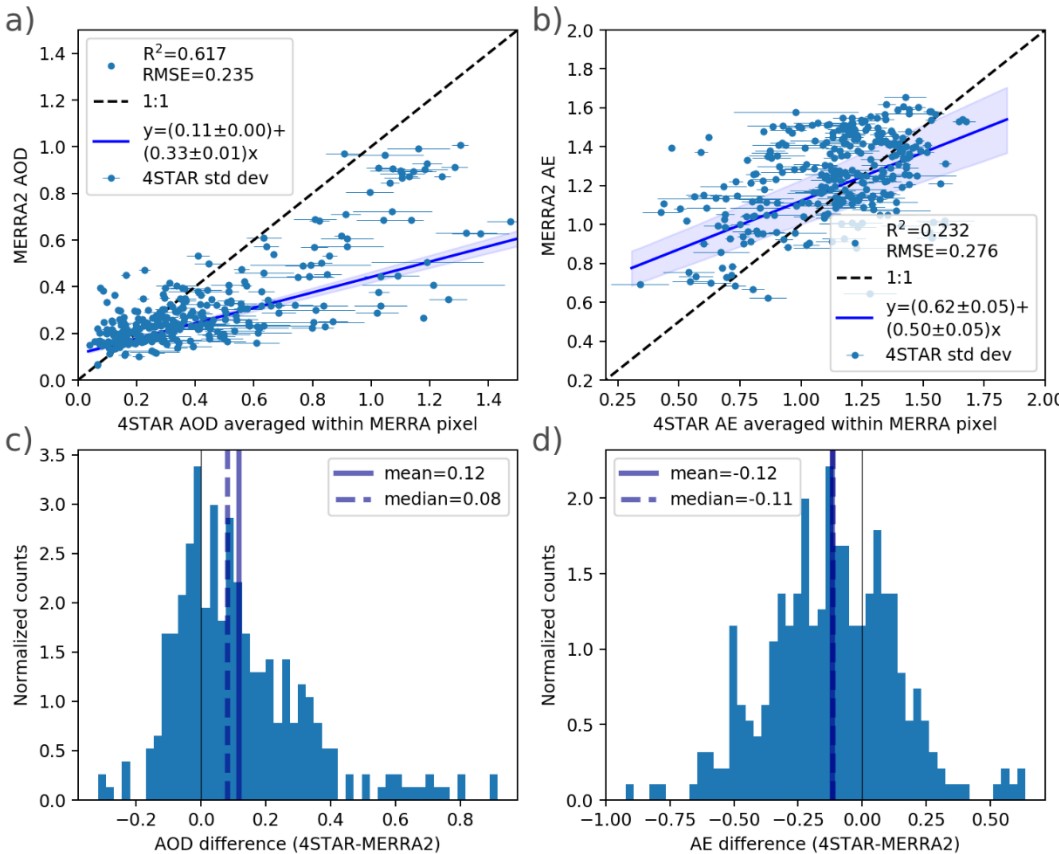

Figure B3 - Comparison of the (a) AOD and (b) AE analysis from MERRA-2 as compared to 4STAR measurements below 500 m. For each pixel from MERRA-2 (at 3 hourly time steps), the nearby 4STAR samples (within 90 minutes of MERRA-2 run) are averaged together for this comparison, with the standard deviation of measured 4STAR AOD within one
MERRA-2 pixel represented by the error bars, shown in the top pair of graphs (a, b). The linear fit used here is the bivariate fit described by York et al. (2004) The difference between AOD and AE from 4STAR and MERRA-2 are shown by the bottom pair of histograms (c, d).

***Code and Data Availability:*** Data for KORUS-AQ is available at the NASA LaRC data archive: https://www-air.larc.nasa.gov/cgi-bin/ArcView/korusaq, following the DOI: 945    10.5067/Suborbital/KORUSAQ/DATA01. The analysis code associated with this work is found at: DOI: https://doi.org/10.5281/zenodo.6965167.

***Author Contributions:*** SEL conceived the study, completed the analysis, and wrote the manuscript. SEL, MSR, JR, CF, collected data and operated 4STAR during KORUS-AQ. SEL, RRJ, SD, and RD worked on technical upgrades to 4STAR. JK, MC created and curated the GOCI AOD products (while MC was at Yonsei). LZ and KLT 950    operated and produced the LARGE products. AD, PC, and QT advised on the use of MERRA-2 aerosol products with AD and PC contributing the 4-D MERRA-2 interpolated data. JR and MK secured ongoing support. All co-authors contributed to edits of the manuscript.

***Competing Interests:*** The authors declare that they have no conflict of interest.

***Acknowledgements:*** NASA DC-8 flight and maintaining staff along with ESPO are thanked for their essential 955    contributions for successful KORUS-AQ deployment. Data collection by John Livingston (retired) is appreciated. NASA Atmospheric Composition Program (ACP) is acknowledged for their ongoing support. GOCI data was provided through "Technology development for Practical Applications of Multi-Satellite data to maritime issues" project of the Ministry of Ocean and Fisheries, Korea.

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
