# Peer review of "Airborne observation during KORUS-AQ show aerosol optical depths are more spatially self-consistent than aerosol intensive properties"

_Atmospheric Chemistry and Physics, 2021_

## Author Comment (AC1)

Response to Reviewers for "Airborne observation during KORUS-AQ show aerosol optical depth are more spatially self-consistent than aerosol intensive properties"

We appreciate comments from Referee #1 (Andrew Sayer) and the push to enhance this manuscript's quality. Please see attached the responses to each comment in blue italic. We have added discussions spanning most of the Referee's comments, with emphasis on descriptions of the expected variability in metrics other than autocorrelation (including a new table) and enhancement to the statistical comparisons using more appropriate bivariate fitting routines. We have also adjusted multiple figures with the Referee's comments in mind. In line with the spirit of the comments, we have enhanced the writing quality throughout the manuscript. See below for more details.

**Referee #1 (Andrew Sayer)**

The topic is important and within scope for ACP. The quality of writing and presentation is high (though I have a few suggestions for changes to Figures). My overall recommendation is for minor revisions. I would be willing to review the revision if the Editor would like. My specific comments and suggestions for revision are as follows:

> *Thank you.*

- The main metric used to quantify the spatial scale of variation is the distance at which the autocorrelation drops off to 85% of its value in the smallest distance bin. I was wondering why 85% was chosen? I would have thought it more common to state in terms of an e-folding distance – unless the autocorrelation profile doesn't look like exponential decay (which some of them might not). Either way I'd appreciate some (brief – not repeating the whole analysis) discussion in the paper of why this particular threshold was chosen and if results qualitatively change if a different metric is used – for example the e-folding distance, or an autocorrelation drop to e.g. 70% of the max rather than 85% (given a correlation of 0.7 corresponds to about 50% of the variance in the field). Looking at curves my guess is in most cases the picture would be the same, but as thresholds are a bit arbitrary it is good to check sensitivity to them.

    > *This is a good question. The purpose of this 85% metric is to catch the inflection point of the autocorrelation, with less uncertainty. We tested multiple different metrics, including the e-folding and at 90%. While the overall trends with respect to decay did not vary much, their magnitude did. We included a new paragraph in section 4.4 and a new table, to describe the distances at various metrics. See here: "The 15% decrease metric is used to identify where there is an inflection point in autocorrelation, however distances where autocorrelation decays by 10% or by 1/e show similar trends (see Table 2 for examples). For AOD, the mean distance where there is 10% decrease in autocorrelation occurs at 26.8 km, roughly 1/2 the distance of 15% decrease, while an 1/e decrease (~37% reduction), the mean distance is 167.5 km. Because of the larger dependence to samples (showing larger spread in autocorrelation at longer distances), the 1/e results in a larger spread of distances (standard*

*deviation of 54 km). Similarly for AE, where the largest standard deviation and spread is found from the 1/e decrease level."*

*And table 2:*

| Relative auto-correlation | AOD$_{500}$ | | | | | AE | | | | |
|---|---|---|---|---|---|---|---|---|---|---|
| | 4STAR | MERRA-2 | GOCI | in situ (LARGE) | **Average** | 4STAR | MERRA-2 | GOCI | in situ (LARGE) | **Average** |
| **90%** | | | | | | | | | | |
| distance [km] | 25 [10, 35] | 65 [35,65] | 10 [3,160] | 7.5 [7.5, 7.5] | **26.88** | 10 [10, 15] | 65 [35, 100] | 0.6 [0.27,1.35] | 0.27 [0.2, 0.27] | **18.34** |
| mean of difference | 0.003 | -0.0015 | 0.008 | 0.013* | **0.003** | -0.0036 | -0.0161 | -0.00098 | 0.0015 | **-0.005** |
| standard deviation | 0.1364 | 0.0504 | 0.0985 | 0.292* | **0.095** | 0.1465 | 0.206 | 0.3072 | 0.1776 | **0.209** |
| **85%** | | | | | | | | | | |
| distance [km] | 35 [25, 35] | 100 [35,100] | 65 [5, 160] | 10 [10,10] | **52.5** | 15 [15,25] | 65 [35,160] | 7.5 [5,7.5] | 0.9 [0.6,0.9] | **22.1** |
| mean of difference | -0.0072 | -0.0073 | 0.0185 | 0.0173* | **0.001** | -0.0039 | -0.0161 | 0.0027 | 0.0048 | **-0.003** |
| standard deviation | 0.1436 | 0.0682 | 0.1442 | 0.318* | **0.119** | 0.162 | 0.2066 | 0.408 | 0.217 | **0.248** |
| **1/e (63%)** | | | | | | | | | | |
| distance [km] | 160 [35,160] | 250 [35,250] | 160 [65, 160] | 100 [35, 100] | **167.5** | 100 [100,100] | 160 [100, 160] | 100 [25, 100] | 65 [65, 100] | **106.25** |
| mean of difference | -0.0073 | -0.0985 | -0.033 | 0.0057* | **-0.046** | 0.037 | -0.065 | 0.028 | 0.037 | **0.009** |
| standard deviation | 0.1557 | 0.1357 | 0.217 | 0.389* | **0.169** | 0.247 | 0.354 | 0.507 | 0.371 | **0.370** |

***Table 2** - Distance bins at which different relative autocorrelation is reached for AOD and AE from 4STAR, MERRA-2, GOCI, and in situ (LARGE). The range in distance (square brackets) is obtained from the autocorrelations that are varied by one standard deviation of the 50-member Monte Carlo ensemble of flight segments. The differences in AOD and AE from all flight segments at the distance bins are reported by their mean and standard deviation. The Average AOD mean and standard deviation of the difference is averaged from 4STAR, MERRA-2, and GOCI, while AE also includes in situ (LARGE). *The in situ extinction coefficient mean difference and standard deviation are multiplied by 2.5 km for easier comparison to the AOD mean and standard deviation values.*

- Related to the above, it would be interesting to quantify at a couple of places what the typical variation in the field is for these autocorrelation drops (e.g. at the distance of 85% autocorrelation, what is the variance of the difference between AOD or FMF or AE at that point and at zero lag). This helps give an idea of how numerically important some of these variations are (with the understanding that these magnitudes might not be transferable to other regions or seasons). For example at the 22.7 km distance where AE autocorrelation has dropped to 85% is the AE difference about 0.1 or 0.3 or?

*To answer this question, and the previous we added a new table which describes the distances at various relative autocorrelation values, and the mean difference in the distance lags. The means are as expected very near zero - implying a somewhat even distribution of the sampling. The standard deviation of the difference between values measured at changing distances does vary, and increase with longer distances. See table 2 above and this new paragraph at the end of section 4.4:*

*"The distances at which autocorrelation varies can also be understood through the expected variation of the aerosol properties (AOD or AE). Since we use the combination of all level flight legs, the difference between AOD or AE between measurements binned by their lag distance has a mean and median very close to zero, while the standard deviation grows with distance. The near-zero mean difference in AOD or AE at varying distances implies an even distribution of measurements. Table 2 showcases the values of the mean, median, and standard deviation for AOD and AE at distances with different autocorrelation reduction."*

- KORUS-AQ also included a dense deployment of ground based AERONET sites (mostly around Seoul). I wonder if these could be used as an additional data source for the autocorrelation analysis to see if the overall picture of relative scales of variation holds as for the 20 DC-8 flights. While they would not be spatiotemporally collocated with the other data sources used, the data have low uncertainty and good temporal sampling. I am not sure if the inter-site spacings are sufficiently varied to fill out the autocorrelation distance profile, but it could be worth looking at the distance pairings to see if this could be a useful addition.

  *While not exactly used for autocorrelation, Choi et al., 2021 examined the distribution of AERONET measurements during KORUS-AQ and the correlation to each other based on distance from different sites. It is more difficult to resolve with stationary measurements, a distance of 50 km roughly translates to a correlation of 85% for fine mode AOD. In contrast the slope of correlation with distance for Fine Mode Fraction is shallower than fine and coarse mode AERONET AOD, suggesting that the aerosol size distribution was the most spatially homogeneous (Fig. 3 in Choi et al., 2021). But the low accuracy of a linear fit to this metric, and the high rate of decrease within the first 100 km of the correlation of one AERONET site to another, suggest a more complex dependence than a linear decrease, as observed in this work (see fig 9). An abbreviated example is presented in section 4.4:*

  *"These results contrast to those reported by ground-based observations from AERONET during KORUS-AQ, as presented by Choi et al., (2021), which shows smaller changes in FMF than AOD for coarse or fine mode as a function of distance between the AERONET ground sites (0.11/100 km for FMF, 0.16/100 km, and 0.14/100 km for AOD fine and coarse mode). However, Choi et al. (2021) also shows a lower correlation in FMF, and arguably, a non-linear relationship, particularly at distances shorter than 100 km. This non-linear relationship is presented here in Fig. 10. "*

  *Reference:*

*Choi, Y., Ghim, Y. S., Rozenhaimer, M. S., Redemann, J., LeBlanc, S. E., Flynn, C. J., Johnson, R. J., Lee, Y., Lee, T., Park, T., Schwarz, J. P., Lamb, K. D. and Perring, A. E.: Temporal and spatial variations of aerosol optical properties over the Korean peninsula during KORUS-AQ, Atmos. Environ., (February), 118301, doi:10.1016/j.atmosenv.2021.118301, 2021.*

- Line 355: the Abstract highlights average and variability of AOD/AE for flights below 500 m but the text here highlights those numbers for flights below 1000 m. Later in the paper there's some discussion of profiles below/above 500 m but the main results here are all framed relative to 1000 m. I thought I'd mention as I'm not sure whether this difference in reporting altitude between the Abstract and main text was intentional.

  *We changed the abstract to reflect the values framed relative to 1000 m to be consistent with the values presented in section 4.1. Other changes throughout the manuscript are intentional.*

- Figures 3, 6: if I understand correctly the spectral plots are means and standard deviations. The data are shown on a log scale so the lower tails of the standard deviations often go down to the y axis. I think it could be more meaningful to plot geometric means (i.e. mean and standard deviation of log(AOD)) or else median and interquartile range (or central 68% of points). These, especially the latter, would be informative of the shape of the AOD distribution at each wavelength.

  *Both figures already showcase the range and interquartile extents for the distribution, but that seems to not be clear. The figure caption and figures (Fig 6 is now Fig 7) themselves have been adjusted and clarified. See here:*
  *"*

[Figure]

*Figure 3 - (a) Histogram of AOD at 501 nm measured by 4STAR distribution from KORUS-AQ, separated by meteorological periods. (b) Corresponding AOD average spectra for each meteorological period, with the error bars denoting the range of AOD (excluding outliers) during that time period, with the thicker bars denoting the interquartile range. The square symbols and error bars are slightly shifted from each other for clarity. The AE in b) is calculated from the average spectra of each respective meteorological period from 453 nm to 870 nm.*

[Figure]

**Figure 7** - *Aggregated AODs observed during KORUS-AQ as a function of observation altitude, for (a) with average AOD spectra, and (b) binned vertically for a subset of wavelengths. The range in binned values are presented by the error bars, while the thicker bar denotes interquartile range (25%-75%). The number of spectra per height bins in a) is 64 736, 41 821, 63 130, 121 569, and 31 076 from lowest to highest respectively. (c) The histogram of the altitude by number of data points (bottom axis) and by number of level legs (top axis), with the mean and median altitudes indicated by solid and dashed lines with the respective colors. "*

- Figures B1, B2, B3 and lines 444-448: I am assuming that the regressions here are ordinary least squares (OLS) linear (unless I missed it, it's not explicit). They should really be removed because this technique is inappropriate for these types of data. Some assumptions required for the validity of OLS linear regression include (a) an underlying linear relationship; (b) independent samples; (c) a single underlying (ideally Gaussian) distribution, (d) negligible uncertainty on the independent variable, and (e) equal variance of the dependent variable across the range of the independent variable. Looking at the clouds of points, the linearity assumption appears invalid for B1(b) and B2(b). The independence assumption is likely invalid throughout given the point of this paper shows high levels of correlation across the domain. The distribution shape assumption is likely invalid since AOD tends to be skewed and closer to lognormal, plus the different meteorological fields having different AOD distributions mean we don't have draws from a single distribution but perhaps 4. The independent variable assumption is valid since, as noted, the 4STAR AOD uncertainty in the midvisible is about 0.03, which is not negligible relative to the low AODs commonly found for the bulk of the data. The AE is also uncertain. Note this assumption can be overcome by use of e.g. reduced major axis (RMA) regression accounting for the uncertainty in the independent variable, but this doesn't help with the others. RMA might also be impractical in the present case because my guess is that a non-negligible fraction of the uncertainty in all the data sets here is systematic (e.g. radiometric calibration uncertainty through deployment) so would also be correlated. The equal variance assumption appears to be violated for panel B3(a) and possibly B1(a) (this can also be overcome using weighted regression if pointwise uncertainties are known beforehand). In short, all the data sets violate some of the

assumptions, and the numbers and uncertainties presented as regression results are not quantitatively correct. The OLS technique is often used in our field but this does not make it right. I recommend the authors remove the regressions from the plots. In any case I don't think they are really needed to get to the main point about the level of comparability of the data. I think showing R2 is ok (as the collinearity of the data is of interest) but rather than regression equations perhaps some metrics like RMS difference, mean offset, mean absolute difference could be used instead. The discussion in lines 444-448 of the paper should be amended as a result. I don't mean to harp on about this point but since inappropriate regressions are common in our field think it's important to try and stop the practice when I get a chance in peer review.

*I've updated the linear fit for all these figures to use a bivariate linear fit, as described by York et al., 2004, and reinforced for use by Cantrell, 2008. These figures also now include the Root-Mean-Square-Error metric for better comparisons and error bars for each x and y coordinate, and are reported in section 4.2. While using this new fit, the corresponding slopes have changed, although the main behavior are unchanged (flatter slope of coarse mode over land than ocean, and better fit for fine mode than coarse mode). The new discussion lines now read:*

*"In addition to the already known AERONET comparisons over land, we find that the coarse mode AOD from GOCI has a lower RMSE over ocean (RMSE=0.093) than land (RMSE=0.112), albeit with relatively low correlation (R2 of 0.058 and 0.066 respectively). This low correlation is accompanied by nearly flat slope when comparing GOCI to 4STAR coarse mode AOD over ocean (0.26±0.05), and less so over land (0.49±0.12), as estimated using a bivariate linear fit (York et al. 2004). The fine mode AOD is much closer to the expected 1:1 line with slopes of 0.83±0.03 and 0.78±0.09 over ocean and land respectively, and low biases of 0.05 and 0.06 for fine mode AOD (see Fig. B2). "*

- Lines 857-859: "Satellite algorithms that assume that aerosol size does not vary as much as aerosol optical depth should be reassessed." I am not aware of data sets produced from algorithms that make assumptions like that, on the scales of tens of km being discussed here – are there any? Most either operate on single pixels (i.e. no spatial constraints) or do multi-pixel processing at a much finer scale than the spatial scales reported on here (e.g. MISR at 4.4 km, GRASP applied to POLDER at 10 km). The VIIRS SOAR ocean algorithm assumes the same fine mode and coarse mode microphysics across 6 km grid cells but AOD and FMF are allowed to vary without spatial constraints for each 750 m pixel within that area. MAIAC used to have some constraints but now retrievals (at 1 km) are spatially independent. It sounds like the GOCI data set used here might (I'm not 100% certain by the way the model selection is described in the paper) but again that's going from 0.5 to 6 km so a lot finer than the scales of variation here. It would be good to either give examples of algorithms here or else delete the comment if there are none using such constraints at the relevant scales.

  *Lines 857-859 has been changed to read: "This work showcases that in some regions the spatial scale at which aerosol size varies is smaller than that for aerosol optical depth."*

Language comments:

- Title: I am not 100% on this, but I am a bit uneasy about "aerosol optical depth are". I think it should either be "aerosol optical depths are" or "aerosol optical depth is".
    *Thank you, changed to "aerosol optical depths are"*

- "Angstrom" should be typeset as "Ångström" throughout.
    *Thank you, changed.*

- Line 428: "sporadic aerosols events" should be "sporadic aerosol events".
    *Thank you, changed.*

---

## Author Comment (AC2)

Response to Reviewers for "Airborne observation during KORUS-AQ show aerosol optical depth are more spatially self-consistent than aerosol intensive properties"

We appreciate comments from the Referee #2 and the push to enhance this manuscript's quality. Please see attached the responses to each comment in blue italic. We have added discussions spanning most of the Referee's comments, with emphasis on the Angstrom exponent uncertainty (including a new section describing a method to calculate the uncertainty, and a new figure) and the statistical comparisons and their importance in terms of observation days and sample number (additional description and new sections to multiple figures). We have also adjusted multiple figures with the comments in mind. In line with the spirit of the Referee's comments, we have enhanced the writing quality throughout the manuscript and added more references to other sources to support our findings. See below for more details.

**Referee #2 (Anonymous)**
General Comments: The authors present a method, based on autocorrelation, to examine the spatial and temporal variability of AOD, AE and FMF over South Korea and adjacent waters during the KORUS field campaign which occurred in May-June 2016. There is much interesting data and analysis to digest in this paper, however there are also some key missing aspects that need further discussion and are very important regarding the conclusions reached.

Some discussion regarding the relative uncertainty of the AOD and AE needs to be added to this manuscript. The AE parameter in general has significant uncertainty due to the individual uncertainties of the spectral AOD that are utilized as input to the calculation. See Kato et al. (2000; JGR) equation 6 and Hamonou et al. (1999; JGR) equations 1-3 for estimates of the uncertainty of AE computations. Also note that the uncertainty of AE increases as AOD decreases while the uncertainty of AOD remains constant for all AOD levels for a sunphotometer such as 4STAR. This is quite important in relation to the analysis presented in this paper, and does not seem to have been considered. Further the range of values of measured AOD and computed AE differ significantly (histograms shown in Figure 4) with the 4STAR values of AE varying over a relatively small range of values ~0.7 to 1.5. The data sample ranges for these parameters coupled with the greater uncertainty in AE relative to AOD also has significant influence on the statistics computed and compared for these parameters and requires further discussion. For these reasons of expected noisier data for AE versus AOD, I have some doubts that the AOD is more spatially consistent than the AE during this KORUS campaign interval data set, at least to the extent suggested. Also the FMF as computed by the SDA algorithm utilizes as the primary input the AE (at 500 nm) and the spectral derivative of AE which has an even greater uncertainty. Therefore the FMF from SDA also has a significant uncertainty (larger than AOD) that also increases at lower AOD levels. Furthermore, the SDA retrieval assumes bi-modality of the aerosol size distribution while in reality three modes may sometimes exist. Specifically in S. Korea the presence of a middle or third mode (of sub- micron radius) from fog processing of sulphate species is often associated with fog over the northeastern Yellow Sea, as documented in Eck et al. (2020) on some of the KORUS flight days.

*Thank you for this comment. We proceeded to add new error propagation analysis as a response. This led to a new figure at the end of section 4.1.*

[revised manuscript text omitted]

*Table 2 - Distance bins at which different relative autocorrelation is reached for AOD and AE for the 4 compared datasets. When the autocorrelations are varied by one standard deviation of the 50-member ensemble Monte Carlo subsampling at 30% of flight segments, we obtain a range of distance bins that relate to the relative autocorrelation and is expressed by the square brackets below the distances. The mean and standard deviation of the differences for the AOD or AE evaluated at the lag distances from all flight segments for the corresponding distance are presented below the distance row for each corresponding relative autocorrelation. The in situ extinction coefficient mean difference and standard deviation are multiplied by 2.5 km for easier comparison to the AOD mean and standard deviation values. The Average AOD and AE columns are the averages from all 4 data sources."*

*The addition of a few sentences in the conclusion are also linked to this uncertainty analysis:*

*"The uncertainty in AE and FMF were also evaluated and shown to peak at low AOD, while the AOD uncertainty was more variable, at larger AODs. "*

*and;*

*"At these same distances at which the autocorrelation is decreased, we find an increase in standard deviation, which is larger than the median uncertainty of AE for the majority of samples. "*

Specific comments:

Line 65: Angstrom needs to be capitalized since it is a proper name.

*Thank you, changed to Ångström throughout the manuscript.*

Line 124: This is misleading as the primary data for this study is from May to mid-June (not May-July as you stated), encompassing a time interval of 41 days.

*Changed July to mid-June.*

Line 136: It could be argued that the economic booms in eastern China and S. Korea and concurrent increases in industrial pollution may have begun a decade earlier than 2010. Please cite more references and evidence for a dramatic increase in fine particle production since 2010.

*Changed since 2010 to "in the preceding decades" And added citations for both China and Korean air quality changes in the recent years (Zeng et al., 2019 and Kim et al., 2020)*

Line 185-185: The decrease in transmittance on the optics window for each flight is a significant source of AOD uncertainty. In Table A1 this decrease in transmittance at 650 nm is given for each flight. However it is well known that deposition of film on optics windows results in transmittance decreases that differ with wavelength. This issue should be discussed as it has a greater direct impact on the Angstrom exponent (in comparison to a single wavelength AOD) since multiple wavelengths are used in the computation of AE.

*The actual computation of the impact of transmittance decrease with film deposition on the optics has been quantified using the full solar spectra from in-flight measurements before and after deposition events. This is clarified in section A.1.3 with the addition of 'The full solar spectra'. Now reading:*

*"The full solar spectra measured at high altitude flight segments (above 6 km) are used to evaluate the change in spectral optical depth due to window deposition, where the AOD is expected to be mostly representative of the stratospheric aerosol, at around 0.03."*

Line 200-201: This is inconsistent with the Section 2.3 title which states Angstrom Exponent is retrieved with GOCI while here in the text you only mention FMF. Which GOCI parameter is actually retrieved and/or used in this section of the paper?

*Changed title of section 2.3 to be 'Fine Mode Fraction'.*

Line 211-212: It would be appropriate and useful for the reader to give the validation/comparison statistics for the GOCI YEAR product of AOD and AE and/or FMF versus AERONET values here in the text.

*We added this summary description, based on the Choi et al., 2018 numbers:*

*"GOCI YAER version 2 has been compared to AERONET measurements over 5 years (Choi et al., 2018), reporting a root mean square error (RMSE) of 0.16 and R=0.91 for AOD over land with N=45 643, and slightly lower (RMSE=0.11, R=0.89) over ocean neighboring AERONET sites (N=18 499). For FMF, the GOCI YAERv2 retrievals are slightly biases towards more coarse mode particles, with comparisons to AERONET for AOD>0.3 having R=0.623 over land. "*

Line 220: Does MERRA-2 really assimilate the AOD product from MODIS or the clear sky radiances from MODIS which are then converted to AOD by an AI algorithm trained with accurate AERONET measurements of AOD? Please check and clarify.

*The description of MERRA-2 assimilation has been changed to specify that MERRA-2 uses radiances directly. Now reads:*

*"In MERRA-2, aerosols are simulated by GEOS model driven by assimilated meteorology fields and assimilates bias-corrected AOD derived from AVHRR and MODIS radiances and AOD from MISR and AErosol RObotic NETwork [...]"*

Line 269-270: It seems that this might be an appropriate place to discuss the flight altitudes of the DC-8 aircraft during this campaign and the lower flight level relative to the faction of AOD below that typical lower flight altitude. At times this paper seems to suggest that total AOD is being investigated but in reality the lowest layers with highest aerosol concentrations are sometimes missing. This is important as much aerosol dynamics (physical and chemical) occurs in the lower boundary layer This issue needs more discussion/clarification in the text.

*We added clarification of the altitude at which these measurements are taken, and Included an estimate of the difference in AOD from 500 m to the surface. This is a new paragraph in section '4.3 Vertical variations in aerosol distribution'. It reads:*

*"The averaged profiles presented in Fig. 7a match individual profiles from the frequent missed-approaches during each flight (3 times per flight near Seoul), the landing and take-off profiles at Osan, and low flight maneuvers over water, particularly for AOD below 500 m. In Fig. 7a and as reported by Choi et al. (2021) when comparing 4STAR to AERONET sites, the difference in AOD from surface to 500 m is roughly 0.1 at 500 nm, but with a highly consistent AE throughout that lower layer. "*

Line 301-302: Please note here in the text how many flight days were utilized, plus how many total flight legs.

*Included the numbers of legs and flights: 304 level legs spanning all 20 research flights.*

Line 307-308: Please be more specific of the distance of these shortest autocorrelation bins.

*Added.*

Line 336-337: It should be discussed here that AERONET requires the 380 nm AOD for L2 retrievals from SDA due to the more robust characterization of alpha' (or curvature; derivative of AE) when utilizing this wavelength. It is surprising that the shortest wavelength considered in your computation of SDA was 452

nm. Also, AERONET uses 5 channels (380, 440, 500, 675, 870 nm) as input to SDA retrievals, not 4 channels as stated here in your text.

> *We corrected the use of standard five wavelengths in AERONET. Additionally we included some expectation of the differences in fine mode fraction calculated not including the shortest wavelength. "Although we used many more wavelengths to characterize the spectral derivative, the shortest wavelength is omitted but is expected to produce results similar to the standard AERONET wavelength set with RMS differences in retrieved fine mode AOD of less than 0.01 (O'Neill et al., 2008)."*

Line 354, Section 4.1: It would be appropriate to make the same type of analysis for AE including maps such as shown in Figure 2 but for AE instead of AOD. Then the spatial variance of the two parameters could be examined at this spatial resolution.

*Added to Figure 2, and modified the caption, and description  in Section 4.1*

[Figure]

*"**Figure 2** - [...]The number of samples is represented by the size of the square symbol for a, b, c, d, e, and f, while the number of days sampled are represented by the size of the circle for g, h, i, j, k, and l. Similarly for AE in spatial bins (g, i, and k), and its standard deviation (h, j, and l)"*

Line 390: What were the minimum, average and median altitudes flown for these transects analyzed in this paper.

*Added a section to figure 6 (now Fig. 7) to address this question, with histogram of altitudes of all measurements and of the level legs. See here, with amended caption*

*"The distribution of altitudes at which each sample is measured and the altitudes where each level legs are similar (Fig. 7c), with some under-representation of level-leg to all samples taken between 3 and 7 km. "*

[Figure]

*"**Figure 7** - Aggregated AODs observed during KORUS-AQ as a function of observation altitude, for (a) with average AOD spectra, and (b) binned vertically for a subset of wavelengths. The range in binned values are presented by the error bars, while the thicker bar denotes interquartile range (25%-75%). The number of spectra per height bins in a) is 64 736, 41 821, 63 130, 121 569, and 31 076 from lowest to highest respectively. (c) The histogram of the altitude by number of data points (bottom axis) and by number of level legs (top axis), with the mean and median altitudes indicated by solid and dashed lines with the respective colors. "*

Line 392-393: The number of days sampled in each grid box would be an important statistic to show since some boxes seem to have small sample size and therefore have a non-representative number of days sampled.

*The number of days sampled in each grid box has been added to the new panels in figure 2, related by the size of the circle symbols. Along with added analysis in section 4.1:*

*"The standard deviation is only weakly correlated to the number of samples within a bin at a Pearson correlation coefficient of $R^2 = 0.13$, with a higher standard deviation for larger number of samples where every additional 403.2±99.9 samples, or 0.9±0.3 days sampled, there is an additional 0.1 standard deviation in AOD. The AE is more dependent on the number of samples or days sampled, with an $R^2 = 0.19$ and $R^2 = 0.36$ respectively, with an increase in standard deviation of AE by 0.1 for locations per 550±111 number of samples, or 2.0±0.3 days sampled. However, this relationship does not seem to hold with the matched MERRA-2 samples, where a higher number of samples or days sampled does not directly translate to higher standard deviation."*

Line 399: The AOD spectra in Fig 3b look somewhat noisy with local minima from ~600 to 625 nm which is also the Chappuis ozone maximum absorption region. I am surprised the AOD spectra do not look

smoother than this in logarithmic coordinates. This has implications for the accuracy of the computed Angstrom Exponent and needs to be discussed in the text.

> *Added a better view of the interquartile range for each of these AODs. We added a sentence in the paragraph: "The AOD in Fig. 3b showcase the spectral dependence over the various periods, particularly with respect to their slope as evaluated over the entire range reported here, which minimize the impact of AOD variations at any one wavelength."*

Line 418-419: Twenty sampling dates although large for an aircraft campaign is not statistically a very large sample size. Plus I suspect there are many fewer days in some parts of the KORUS domain shown in Figure 2.

> *These are showcasing not only singular location in those days, but a path over Korea, this section tries to tackle the representativeness of the airborne observations for the broader region by comparisons to GOCI and MERRA-2. While 20 days may not be statistically significant, this manuscript offers a connection to satellite and model, which can in a future study be used to expand in time.*

Line 442-444: This is clearly an exaggeration to say that 5.8% of the variance explained is less than 6.6% of the explained variance, in a statistical sense. They are essentially equal for all practical purposes, within less than 1%.

> *To answer both these concerns and Reviewer #1, we updated Figure B2, and the metrics associated with them, by including RMSE calculations that show more differences between land and ocean.*
> *This portion now reads: "In addition to the already known AERONET comparisons over land, we find that the coarse mode AOD from GOCI has a lower root square mean error (RMSE) over ocean (RMSE=0.093) than land (RMSE=0.112), albeit with relatively low correlation (R2 of 0.058 and 0.066 respectively)."*

Line 490-492: This is significant, especially if the level flight legs miss a significant portion of the total column AOD due to restrictions on the minimum flight altitude. The portion of the total column AOD that are missed by the lowest flight legs needs to be estimated and documented in the text of this manuscript.

> *The lowest portion of the atmosphere, down to the surface, have been sampled multiple times throughout each flight by missed-approaches, landing and take-off, and low flight legs over water. We added this passage to the paragraph to address this link:*
>
> *"The averaged profiles presented in Fig. 6a match individual profiles from the frequent missed-approaches during each flight (3 times per flight near Seoul), the landing and take-off profiles at Osan, and low flight maneuvers over water, particularly for AOD below 500 m. In Fig. 6a and as reported by Choi et al. (2021) when comparing 4STAR to AERONET sites, the difference in AOD from surface to 500 m is roughly 0.1 at 500 nm, but with a highly consistent AE throughout that lower layer. "*

Line 494-497: This is also important, as the upper layer has a higher coarse mode fraction and is likely more homogeneous in AOD due to mixing and dispersion in time and space from distant dust source regions.

> *Yes, we agree, thus the presentation of this caveat.*

Line 534-537: The AE is lower below 2 km during the extreme pollution/transport regime due to larger size fine mode particles for those dates, see Table 1 in Eck et al. 2020 for the large fine mode radius during this time period. Therefore I think you are mistaken to identify this as greater coarse mode influence due to lower AE. The FMF from SDA should still be quite high for this extreme pollution/transport period. This highlights an issue with the AE parameter (as computed from the wavelengths you used) since it is not always indicative of fine/coarse mode relative influence. The aerosol during the extreme pollution/transport period is affected by humidification plus fog and/or cloud processing during this high cloud fraction and high humidity time period.

> *Yes, you are right and this is an error. We looked at the Fine mode fraction vertical distribution for this time period. While not substantively different from the AE figure in the manuscript, the extreme pollution below 2 km does have a much higher fine mode fraction than anticipated. See the figure below. We corrected the statement, which now reads:*
>
> *"The extreme pollution / transport regime shows a stratification of the aerosol layer for small AEs at lower altitudes (below 2 km) than all the other periods, however the FMF for that same time period and vertical region is highest than all other observations. This supports the observations by Eck et al. (2020) that a larger peak of fine mode is present during this period, relating to growth of small particles due to humidification or cloud-processing. "*

[Figure]

Line 558: Fine mode aerosols are never subjected to 'only transport' as you suggest here, since aging processes such as coagulation occur, and condensation occur during transport plus possible interaction with clouds/fog and particle humidification. The fine mode dominated in all but one flight day.

*To answer this and the point presented by the other reviewer, we changed the wording of this section to read:*

*"When considering aerosol transport, the intensive properties are expected to remain constant, such as size, mass absorption efficiency, and index of refraction, but the total concentration within a column, impacting the AOD, would change due to dilution and removal of the aerosol (e.g. via rain out or dry deposition). For example, dust aerosol transported from mainland China, which after initial growth with chemical and morphology changes, by coagulation and condensation, near source or after cloud/fog processing and humidification/dehumidification, have near constant intensive properties but experience dilution causing reduction in AOD, but no change in spectral dependence. "*

Line 573-574: I disagree that this would only occur within a small distance. After long distance transport, fine mode properties can be significantly modified by cloud/fog processing within droplets with subsequent droplet evaporation yielding different particle properties.

*Yes, this is an omission in this sentence. We have changed it to read:*

*"For example, dust aerosol transported from mainland China, which after initial growth with chemical and morphology changes, by coagulation and condensation, near source or after*

*cloud/fog processing and humidification/dehumidification, have near constant intensive properties but experience dilution causing reduction in AOD, but no change in spectral dependence. "*

Line 575-576: What is the average flight altitude of these data in Figure 9? Please give the mean, median, minimum and maximum altitude as this is pertinent to the informed evaluation of these results.

*To answer this and the previous point we included a histogram of the altitudes at which we measured all data and for the level flight legs. See the amended figure 6 (now Fig 7).*

Line 593: This is not really total column AOD from 4STAR. How much of the AOD is below the flight altitude it unclear as this has not been adequately discussed in the manuscript.

*We removed 'total column' from this sentence. For estimates of the adjustment to ground and flight level, see the newly added paragraph in section 4.3, (from response above).*

Line 684-685: Please state how many days of 4STAR data were utilized in each of the KORUS meteorological periods. I suspect that the sample size in terms of number of days is not very robust in most of these periods.

*We added the number of flights for each period in table 1, which summarizes the meteorological periods.*

Line 712-715: There was also some sea breeze pushing back and forth of aerosol over from the Yellow Sea to over land (and vice versa) during the stagnation period, not just local effects of aerosol production and evolution. Additionally, this period had the lowest AOD and therefore the largest uncertainty in AE and FMF. In fact the uncertainty in AOD approaches the AOD magnitude in the long wavelength visible and NIR during the stagnation period.

*We added these observations as discussion, which reads: "In tandem, the ebb and flow of sea breeze shifted aerosol from over the yellow sea and Seoul, likely contributing to variations in aerosol during this time period (Eck et al., 2020)."*

*In addition to a caveat regarding the uncertainty in AE and FMF "[...], albeit with higher uncertainty in FMF and AE measurements due to low AOD"*

Line 734-735: There was no evidence of significant dust transport during the extreme pollution transport period as you seem to imply here. The total column FMF is very high from AERONET data using SDA for this meteorological period. Please provide evidence of this dust if you have it since no other published KORUS paper had documented that phenomenon for this particular meteorological period.

*Added in addition to the data presented here, reference to the DIAL/HSRL data identifying a portion of the aerosol layer as being 'dust mix'. These lines now read:*

*"Additionally, these aerosols were identified as dust mix type because of their high depolarization ratio in the 3 km to 7 km range using the DIAL (Differential Absorption Lidar) / HSRL (High Spectral Resolution Lidar) measurements for that time period (KORUS-AQ Science Team 2019; HSRL - DIAL KORUS-AQ Flight 19 - May 30, 2016, accessed 2022). "*

*References: HSRL - DIAL KORUS-AQ Flight 19 - May 30, 2016,*
*https://science-data.larc.nasa.gov/lidar/korus-aq/, last accessed 28 June 2022.*

Line 735-737: Also it should be noted that this transport period had the highest AOD and therefore the smallest uncertainties in both the AE and FMF. This period has very large fine mode particles related to the high cloud fractions and high RH therefore strongly suggesting particle humidification and/or cloud/fog droplet processing.

*Agreed. We added this portion of sentence to reflect these observations:  "[...] and large fine mode particles occurring during high RH and cloud fractions suggesting cloud processing or particle growth by humidification (Eck et al., 2020)"*

Line 737-738: Note that there is evidence for new particle formation in the extreme pollution transport period. See the increased PM2.5 in central Seoul versus the west coast during this interval (Eck et al., 2020). However your flight lines may not be able to identify this phenomenon as it is manifested in surface PM, possibly not at the flight altitudes of the flight segments which were analyzed in this study.

*Without having a detailed measurement of the evolution of the aerosol over all space and time, it is hard to confirm new particle formation. The increase in PM 2.5 at the ground could also be linked to variations in wind patterns, not uniquely sea-breeze or land-breeze. That said, we revised our statement to be a bit more exact. It is not the new particle formation, but rather inhomogeneous in space and time new particle formation that is not observed. It now reads:*

*"[...] e.g. no large spatial variations in rain out, new particle formation, or particle growth. There are increased ground-based observations of the small PM2.5 particles potentially linked to new-particle formation (e.g., Eck et al., 2020), however these are either not observed here, or not varying the FMF as much as the AOD. "*

Line 762: Please give the number of days of sampling for each of the three altitude layers.

*Added in parenthesis : "(greater than 3 km altitude, from all 20 flights)  [...] (1-3 km, sampled by 18 of the 20 flights) [...]  (below 1 km, sampled by 18 of the 20 flights)"*

Line 832-833: This is an odd emphasis in the Conclusions section on dust at higher altitudes as the total column AOD during the extreme pollution period was dominated by fine mode aerosols.

*Quite true, but given the already largely studied fine mode fraction, it seems important to highlight this finding - particularly not observable from the ground.*